# SCALING LAWS FOR GENERATIVE REWARD MODELS

## ABSTRACT

We study the scaling behavior of generative reward models (GenRMs) for reinforcement learning from AI feedback (RLAIF) when used as drop-in replacements for Bradley-Terry models to optimize policies. Building on established scaling laws for reward model overoptimization, we investigate whether GenRMs, particularly those employing chain-of-thought reasoning, exhibit different robustness properties as policies drift from their training distribution during gradient updates. Using the Qwen3 model family (0.6B–14B), our study includes systematic evaluation of thinking GenRMs (trained via GRPO) against answer-only variants (trained via SFT) across policy size, reward model size, reward model type, training budget, and the $\beta$ parameter in online DPO. Our results reveal a consistent *evaluator-rewarder gap*: thinking GenRMs outperform answer-only variants by 1–2% on validation tasks, yet these gains diminish—and often reverse—during policy optimization, where answer-only GenRMs achieve higher Gold Elo and more stable proxy-Gold alignment. We find that reward model scale is the most decisive factor for policy quality, with gains continuing even when the GenRM far exceeds the policy in parameters. Moreover, intermediate GRPO checkpoints of thinking judges can outperform fully-trained checkpoints as rewarders, despite worse static accuracy. We track these dynamics with Elo arenas under both proxy and Gold evaluation, providing a fine-grained proxy–Gold alignment diagnostic beyond saturated validation metrics.

## 1 INTRODUCTION

Policies trained via reinforcement learning from human feedback systematically exploit imperfections in their reward models, degrading true quality even as proxy metrics improve (Gao et al., 2022; Rafailov et al., 2024). Generative reward models (GenRMs) have emerged as a promising response: by formulating evaluation as conditional text generation rather than scalar prediction, GenRMs can leverage pre-training priors and chain-of-thought reasoning and achieve strong performance on static preference benchmarks (Mahan et al., 2024; Zhang et al., 2025; Whitehouse et al., 2025). This success has fostered an implicit assumption that more accurate static evaluators should translate to more effective online rewarders.

Yet GenRMs have been studied almost exclusively as judges on fixed distributions, not as in-loop rewarders where the policy distribution shifts during training. In this work, we test that assumption and target the practitioner-facing question it conceals: *if one chooses a GenRM as the reward signal, how should compute and architecture be allocated to maximize downstream policy quality?* We study three design dimensions in the online setting where they matter: (i) thinking versus answer-only judging, (ii) judge scale, and (iii) policy scale to maximize downstream policy quality rather than static accuracy? We instantiate these axes in creative writing using LITBENCH (Fein et al., 2025), a non-verifiable preference domain where correctness cannot be programmatically checked. In such domains, static evaluator accuracy can saturate while reward hacking and proxy-Gold divergence remain salient, making creative writing an ideal testbed for whether stronger static evaluators translate to better in-loop reward signals.

**Contributions.** (1) We present the first systematic scaling study of GenRMs deployed as online reward signals, sweeping judge format, judge size (0.6B–14B), policy size (0.6B–8B) and training budgets under online DPO. (2) We introduce a proxy–Gold Elo diagnostic that measures alignment under optimization pressure and supports cross-configuration comparison. (3) We provide empirical

guidance for GenRM deployment: judge scale and training duration dominate, answer-only Gen-RMs yield more stable optimization than thinking variants, and intermediate GenRM checkpoints can outperform fully-trained models as rewarders.

Our findings reveal a consistent *evaluator–rewarder gap*. Thinking GenRMs improve in-distribution accuracy by 1–2% across all scales, yet these gains diminish—and often reverse—when the same judges drive online optimization. Answer-only GenRMs yield higher peak Gold Elo and more stable proxy–Gold alignment. In contrast, scaling judge capacity proves decisive: larger judges continue improving policy quality even when far exceeding the policy in parameters, suggesting that judging is at least as hard as generating in this domain.

## 2 METHODOLOGY

We begin with a human preference dataset of the form

$$\mathcal{D}_{\text{human}} = \{x^{(i)}, y_A^{(i)}, y_B^{(i)}, I_H^{(i)}\}_{i=1}^N,$$

where $x^{(i)} \in \mathcal{X}$ are prompts, $y_A^{(i)}, y_B^{(i)} \in \mathcal{Y}$ are pairs of responses to the prompts, and $I_H^{(i)} \in \{A, B\}$ denotes the human-preferred response.

**Gold evaluator.** Since human preferences are not available on demand during policy training, we cannot directly query them to measure policy performance at each checkpoint. To enable continuous evaluation throughout the optimization process, we train a large generative reward model (GenRM), denoted $V_{\text{gold}}$, on $\mathcal{D}_{\text{human}}$. This model serves as a *Gold* evaluator that provides a stable, queryable preference distribution anchored to human judgments. Grounding $V_{\text{gold}}$ in human preferences ensures that our experimental setup reflects real-world alignment scenarios where the ultimate objective is to match human judgment. However, our study does not require $V_{\text{gold}}$ to perfectly replicate human preferences. Rather, we require a *consistent anchor distribution* against which we can measure proxy GenRM behavior as policies drift during training. The central question we investigate is how well smaller proxy GenRMs generalize when used as reward signals, relative to this anchor—not whether the anchor itself is perfectly human-aligned.

To make evaluation computationally feasible, we choose $V_{\text{gold}}$ to be an *Answer-Only* model that outputs a single indicator token. Using $V_{\text{gold}}$, we construct a Gold Preference dataset:

$$\mathcal{D}_{\text{gold}} = \{x^{(i)}, y_A^{(i)}, y_B^{(i)}, I_G^{(i)}\}_{i=1}^M,$$

which is then used to train a variety of smaller GenRMs.

**Proxy GenRMs.** We consider two types of GenRMs: *Answer-Only* and *Thinking*. The Answer-Only models output a direct judgment token ($A$ or $B$):

$$I \sim v_{\text{ans}}(\cdot \mid x, y_A, y_B),$$

while the Thinking models first generate a reasoning trace $z$ before producing the final verdict:

$$z \sim v_{\text{think}}(\cdot \mid x, y_A, y_B), \quad I \sim v_{\text{think}}(\cdot \mid x, y_A, y_B, z).$$

**Training objectives.** We train Answer-Only models using supervised fine-tuning (SFT) and Thinking models using GRPO. This asymmetry reflects the training pipelines used by practitioners in real-world settings. For Answer-Only models, the supervision signal is unambiguous: given a preference label, the model should predict the corresponding token. SFT directly optimizes this objective and is the natural choice. For Thinking models, however, the supervision problem is fundamentally different. A chain-of-thought has no single correct trajectory, and current practice overwhelmingly relies on reinforcement learning objectives like GRPO. Our goal is not to perform an architectural ablation under identical training algorithms, but rather to compare the best-supported pipeline for each model type—training each with the method that empirically performs best for its structure.

For GRPO, we employ two reward signals:

1. **Accuracy reward:** A binary reward (1 if the model reaches the correct verdict, 0 otherwise).

2. **Positional consistency reward $r_{\text{pos}}$:** Inspired by prior work, we observe that models often produce contradictory judgments when the order of $(y_A, y_B)$ is swapped in the prompt. To mitigate this, we explicitly place both orderings $(A, B)$ and $(B, A)$ into the *same GRPO group*, and compute a group-level majority vote over the sampled completions. A reward of 1 is assigned only if the majority verdicts under both orderings are consistent and match the correct label. To avoid introducing noise, $r_{\text{pos}}$ is only given to completions that end at the correct verdict.

We refer to these trained GenRMs as *Proxy models*, and evaluate them with respect to $V_{\text{gold}}$ preferences.

**Policy optimization.** We train policies $\pi_\theta$ of varying sizes using *Online Direct Preference Optimization* (Online DPO). DPO is a natural choice for preference-based policy optimization because GenRMs produce pairwise preferences as their native output, which DPO can consume directly without requiring scalarization or additional transformations. In *offline* DPO, response pairs are sampled once from a fixed reference policy and reused across many gradient updates, keeping the supervision distribution static throughout training. In *online* DPO, response pairs are repeatedly resampled from the current policy after each update, so the supervision distribution evolves as the policy changes. This online setting is particularly important for our study, as we aim to understand GenRM robustness under distribution shift induced by policy optimization. Empirically, online preference optimization has been shown to yield more stable and effective alignment dynamics when implemented carefully (Guo et al., 2024).

The policies are trained on prompts sampled from the same distribution $\mathcal{X}$ used for $V_{\text{gold}}$ and the Proxy models. During training, we periodically save checkpoints and sample responses on a fixed validation set of prompts. To evaluate these checkpoints, we compute ELO ratings based on pairwise comparisons of their responses. ELO evaluation provides a more fine-grained measurement than raw win rates against a fixed reference distribution. We compute ELOs with respect to both Proxy models and the Gold model, enabling us to analyze the relationship between Proxy-based evaluation and Gold-standard evaluation.

# 3 EXPERIMENTAL SETUP

**Dataset construction.** We construct three disjoint datasets for our experiments:

- **Human preference dataset $\mathcal{D}_{\text{human}}$:** We sample 21,000 preference triplets $(x, y_A, y_B, I_H)$ from LITBENCH Fein et al. (2025), a large-scale preference dataset over human-written stories from Reddit. We split these into 20,000 for training and 1,000 for validation. This dataset is used exclusively for training the Gold evaluator.

- **Gold preference dataset $\mathcal{D}_{\text{gold}}$:** We sample a separate, non-overlapping set of 21,000 triplets from LITBENCH and re-annotate them using the trained Gold evaluator, yielding $(x, y_A, y_B, I_G)$ triplets. Critically, while both $\mathcal{D}_{\text{human}}$ and $\mathcal{D}_{\text{gold}}$ are sourced from LITBENCH, they represent different preference distributions: the former reflects human judgments, while the latter reflects the Gold model's judgments. We split $\mathcal{D}_{\text{gold}}$ into 20,000 for training and 1,000 for validation. This dataset is used for training and evaluating Proxy GenRMs.

- **Policy training dataset:** We source prompts from the WRITINGPROMPTS dataset[1], the ancestor dataset from which LITBENCH was constructed. To avoid data leakage, we exclude all prompts that appear in LITBENCH, yielding 199,000 candidate prompts. We randomly sample 91,000 prompts (90,000 for training and 1,000 for validation) unless specified otherwise in specific experiments.

Building on prior work that has extensively evaluated various reward models on LITBENCH facilitates future research and cross-comparisons between our results and other studies.

**Model families.** We use the QWEN3 model family for all GenRMs and policies. For Proxy GenRMs, QWEN3 is the only open-weight model family that supports both Thinking and Answer-Only

---

[1] https://huggingface.co/datasets/euclaise/WritingPrompts_preferences

modes across multiple parameter sizes. This enables controlled comparisons where we can vary GenRM mode and scale while holding the backbone architecture and instruction-tuning distribution fixed. For policy models, using the same backbone as the GenRMs allows us to cleanly isolate the interaction between policy size and GenRM size, which is central to our scaling analysis.

**Gold GenRM training.** We train the Gold evaluator from QWEN3-32B using supervised fine-tuning on $\mathcal{D}_{\text{human}}$. To improve positional robustness, we train on both orderings $(y_A, y_B)$ and $(y_B, y_A)$, resulting in 40,000 training pairs. We use a batch size of 128 and a learning rate of $1 \times 10^{-5}$.

**Proxy GenRM training.** We train Proxy GenRMs from the QWEN3 series at sizes {0.6B, 1.7B, 4B, 8B, 14B} using $\mathcal{D}_{\text{gold}}$. We refer to the off-the-shelf, untrained QWEN3 models as `Baseline-{size}-Ans` and `Baseline-{size}-Think` for Answer-Only and Thinking modes, respectively. For Answer-Only models, we use supervised fine-tuning with batch size 128 and learning rate $1 \times 10^{-5}$. As with the Gold model, we train on both positional orders, yielding 40,000 training pairs. For Thinking models, we use GRPO with the following configuration: 16 prompts per step, group size 8, minibatch size 64, no KL penalty, and learning rate $1 \times 10^{-6}$. We train on both positional orders. We refer to the trained Proxy GenRMs as `GenRM-{size}-Ans` for Answer-Only models and `GenRM-{size}-Think` for Thinking models, where {size} denotes the parameter scale.

**Policy training.** We train policies $\pi_\theta$ from the QWEN3 series at sizes {0.6B, 1.7B, 4B, 8B} using Online DPO. All policies are trained with thinking disabled. We use a training batch size of 192, minibatch size of 64, learning rate of $1 \times 10^{-6}$, and $\beta = 0.02$ unless specified otherwise in specific experiments. For each sampled prompt, we generate response pairs from the current policy and evaluate them using a Proxy GenRM. To ensure positional consistency, we compute preferences under both orderings $(y_A, y_B)$ and $(y_B, y_A)$, retaining only pairs where the two orderings agree. We restrict prompt length to 512 tokens and response length to 2048 tokens.

**Evaluation.** During policy training, we save checkpoints every 10 training steps. For each checkpoint, we sample responses on the fixed validation set and compute ELO ratings based on pairwise comparisons judged by both the Proxy GenRM used during training and the Gold evaluator. Policy training runs for 460 steps in the main experiments (yielding 47 checkpoints), though specific ablations may use different training durations.

**Reproducibility and LLM usage.** Beyond the GenRM and policy models described above, we did not use LLMs for dataset filtering, rubric creation, label consolidation, or prompt generation. All prompts used for policy generation and GenRM pairwise evaluation are provided in Appendix B.

## 4 EXPERIMENTS AND RESULTS

### 4.1 GENRM TRAINING

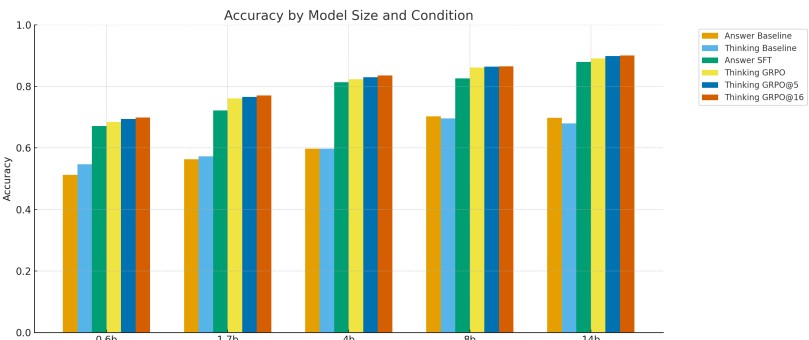

Figure 1: Performance of trained GenRMs of different sizes and training methods on the Gold Preference dataset. Thinking GenRMs consistently outperform Answer-Only models across all scales, while both substantially exceed baseline performance. **Takeaway:** Training yields ~20% accuracy gains; Thinking models provide a consistent 1–2% edge as static evaluators.

**Gold evaluator alignment.** The Gold GenRM achieves 79% agreement with human preferences on the held-out validation split of $\mathcal{D}_{\text{human}}$. Prior work on LITBENCH reports peak accuracies of approximately 78% Fein et al. (2025), indicating that our Gold evaluator achieves strong alignment with the human preference distribution. While our study only requires a stable reference distribution rather than perfect human alignment, the 79% agreement demonstrates that the Gold evaluator is well-grounded in human judgments.

**Trained GenRMs substantially outperform baselines.** We train GenRMs of sizes 0.6B, 1.7B, 4B, 8B, and 14B, with both Answer-Only (SFT) and Thinking (GRPO) variants (Figure 1). The untrained baseline models (both Answer-Only and Thinking) achieve similar accuracies across all sizes, suggesting no apparent bias in the Gold evaluator toward either format. Across all scales, trained models show an approximately 20% accuracy improvement over their respective baselines, confirming substantial room for optimization and indicating that the Gold anchor distribution is sufficiently distinct from the untrained proxy distributions.

**GRPO is an effective training method for Thinking GenRMs.** To validate our choice of GRPO for training Thinking models, we conducted an ablation study on the 4B model. When evaluated on the Gold Preference dataset, the off-the-shelf model achieves 60.0%, STaR achieves 60.7%, distillation from QWEN3-32B thinking traces (SFT-style) achieves 64.7%, and GRPO with accuracy and positional consistency rewards achieves 83.2%. Only GRPO yields substantial improvements, validating its use as the primary training method for Thinking GenRMs.

**Thinking GenRMs outperform Answer-Only models on in-distribution evaluation.** On average across all scales, trained Thinking GenRMs achieve 1.7% higher accuracy than their Answer-Only counterparts when evaluated on the Gold Preference dataset. To evaluate whether sampling multiple completions yields additional gains, we compute majority-vote accuracy over $k = 5$ and $k = 16$ samples from the Thinking models (Figure 1). This yields modest further improvements of 0.5% and 0.8% respectively (e.g., at 14B, 89.1% → 89.6% → 90.0%). Given the computational overhead and minimal gains, we do not employ multi-sampling during policy training.

## 4.2 TRAINED GENRMS VS. BASELINE MODELS IN POLICY TRAINING

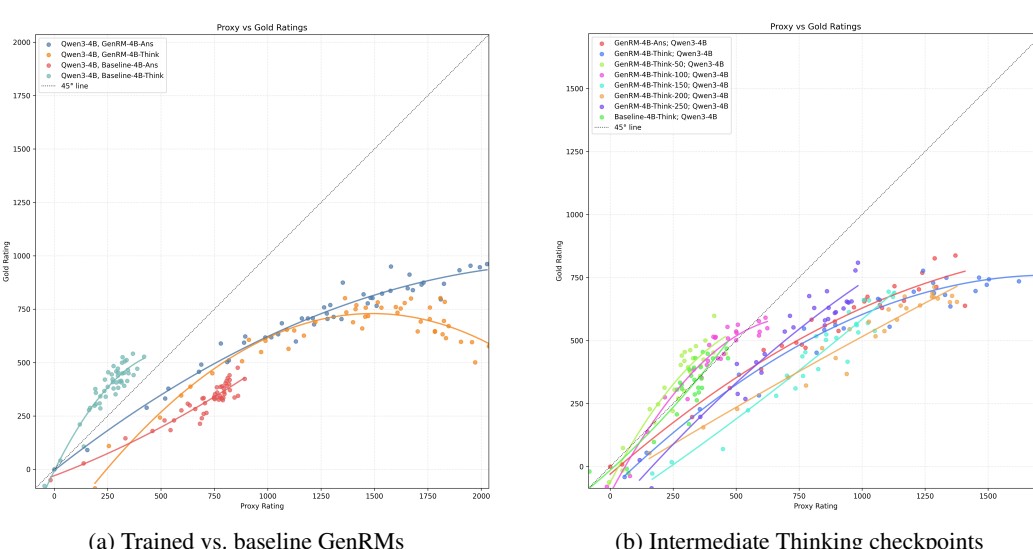

(a) Trained vs. baseline GenRMs      (b) Intermediate Thinking checkpoints

Figure 2: Proxy vs. Gold ELO ratings for policies trained with different GenRMs. (a) Comparison of trained vs. baseline GenRMs. (b) Policies trained on 50,000 prompts with Thinking GenRMs, where the GenRM is checkpointed at intermediate stages of GRPO training (50–250 steps). In both plots, the x-axis shows Proxy ELO and the y-axis shows Gold ELO, with the dotted line indicating perfect alignment ($45°$).

*Does training the GenRM improve downstream policy quality compared to off-the-shelf models?*

We compare policy training using trained versus baseline (off-the-shelf) GenRMs. In this experiment, both the policy and GenRM are 4B models, with Answer-Only and Thinking vari-

ants (Figure 2a). Trained GenRMs dramatically outperform their baseline counterparts. Notably, while Thinking outperforms Answer-Only at baseline, this advantage reverses after training: `GenRM-4B-Ans` exceeds `GenRM-4B-Think` by over 200 Elo points. We discuss this reversal further in the next subsection.

## 4.3 ANSWER-ONLY VS THINKING

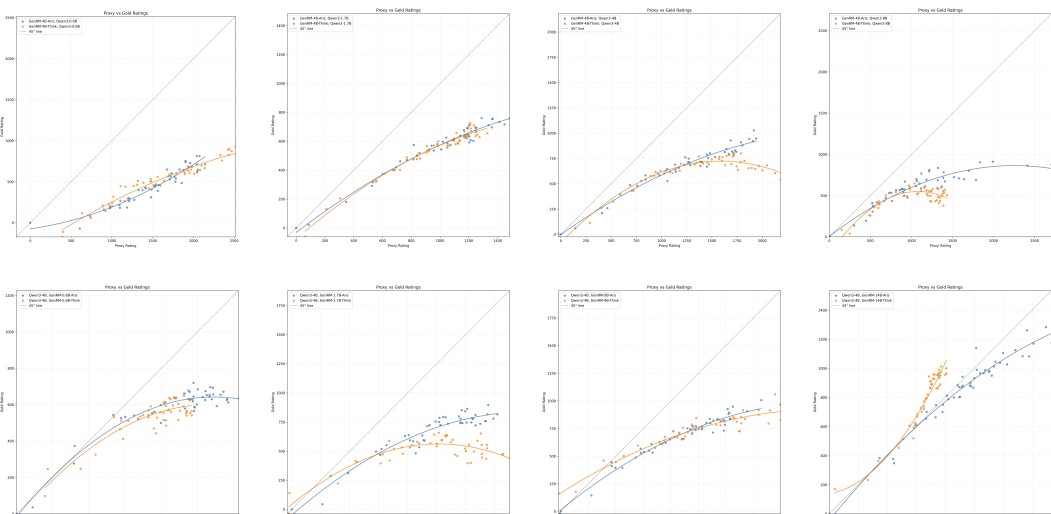

Figure 3: Proxy vs. Gold ELO ratings for policies trained with Answer-Only (blue) vs. Thinking GenRMs (orange) across multiple policy sizes and GenRM sizes. Top row: policy sizes 0.6B, 1.7B, 4B, 8B with fixed 4B GenRM. Bottom row: 4B policy with GenRM sizes 0.6B, 1.7B, 8B, 14B. The dotted line indicates perfect alignment. **Takeaway:** Answer-Only GenRMs consistently achieve higher Gold Elo and tighter proxy-Gold alignment than Thinking GenRMs across all configurations.

*Does chain-of-thought reasoning improve GenRM effectiveness as online rewarders?*

Across all combinations of policy sizes (0.6B, 1.7B, 4B, 8B) and GenRM sizes (0.6B–14B), policies trained with Answer-Only GenRMs achieve both higher maximum Gold ELO and smaller discrepancies between Proxy and Gold ratings compared to those trained with Thinking GenRMs (Figure 3).

This finding is surprising for two reasons. First, in-distribution evaluation (Section 4.1) showed Thinking GenRMs outperforming Answer-Only models by ~1–2% accuracy. Second, baseline (untrained) models displayed the opposite trend, with Thinking variants aligning better with the Gold model than Answer-Only. Despite these initial advantages, Thinking GenRMs prove less reliable when used as reward models for online policy optimization.

We interpret this as evidence that Thinking GenRMs are more vulnerable to off-distribution shifts introduced by policy training. Two factors may explain this gap.

First, higher in-distribution GenRM accuracy does not necessarily translate to better policy training. Since Online DPO changes the response distribution during training, GenRM robustness across the full domain matters more than performance on a fixed validation set. Figure 2b provides direct evidence: we trained policies using intermediate checkpoints of a 4B Thinking GenRM during GRPO training (at 0, 50, 100, 150, 200, and 250 steps). While GenRM accuracy steadily improves on the in-distribution set (from 59.7% to 82.3%), policy Gold Elo peaks at an intermediate checkpoint and then degrades. Notably, the 250-step checkpoint (79.2% accuracy) produces policies that perform close to the Answer-Only baseline in Gold Elo, with a more favorable slope between Proxy and Gold ratings. This suggests overfitting to the evaluation distribution hurts generalization during policy optimization, and that intermediate Thinking checkpoints may provide more effective learning signals than fully-trained models.

Second, GRPO optimizes final preferences rather than reasoning correctness, allowing flawed reasoning paths that still reach correct labels. These weaknesses may be tolerated in-distribution but

exploited out-of-distribution. Because Thinking models split computation between latent activations and explicit language traces, this may introduce additional failure modes compared to Answer-Only models, which perform computation entirely in the latent space.

In summary, Answer-Only reward models are more token efficient and demonstrably more robust for preference-based policy optimization in this domain, though intermediate Thinking checkpoints retain potential if their training dynamics are better understood.

## 4.4 EFFECT OF GenRM SIZE

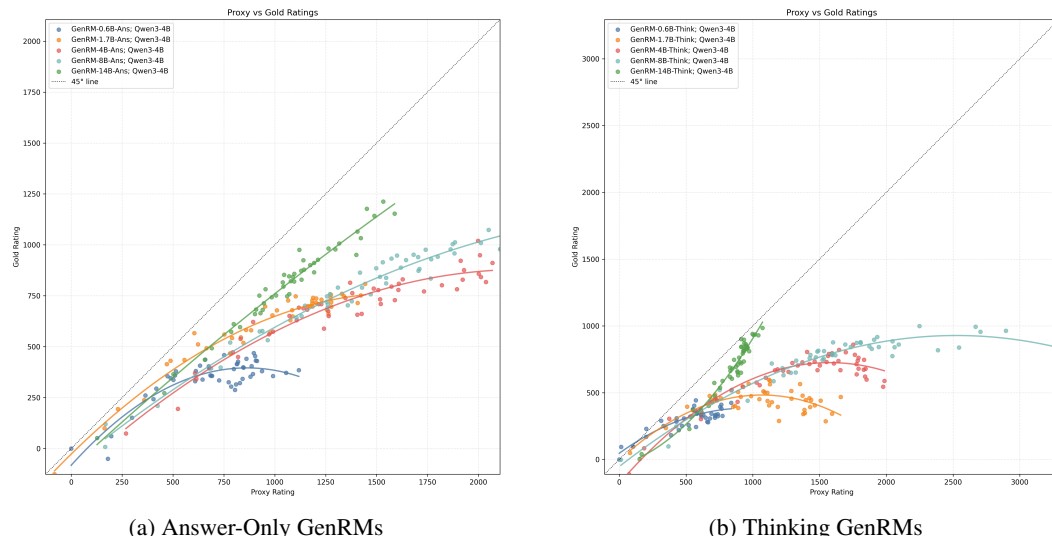

(a) Answer-Only GenRMs                           (b) Thinking GenRMs

Figure 4: Proxy vs. Gold ELO ratings for policies trained with GenRMs of different sizes. The policy is fixed at 4B. Left: Answer-Only GenRMs (0.6B–14B). Right: Thinking GenRMs (0.6B–14B). The dotted line indicates perfect alignment. **Takeaway:** Increasing GenRM size improves Gold Elo and tightens proxy-Gold alignment, with gains continuing even when the GenRM far exceeds the policy in parameters.

*Does scaling the reward model improve policy quality, and do gains continue when the GenRM exceeds the policy in size?*

We investigate the effect of scaling GenRM size while fixing the policy size at 4B. Figure 4 shows results for both Answer-Only and Thinking GenRMs across scales from 0.6B to 14B.

We observe clear and consistent gains from increasing GenRM size. Notably, performance continues to improve even when the GenRM is much larger than the policy: at 14B, both Answer-Only and Thinking models yield substantial gains over smaller counterparts. This trend suggests that the capacity of the evaluator plays a decisive role in stabilizing and guiding preference-based training. The result highlights an important asymmetry: within this domain, "judging" appears to be as hard—or harder—than "generating."

## 4.5 EFFECT ON POLICY SIZE

*Does scaling the policy improve quality, and does the effect differ between Answer-Only and Thinking supervision?*

We examine the effect of scaling policy size while fixing the GenRM. Figure 5 shows results for policies ranging from 0.6B to 8B trained with both Answer-Only and Thinking GenRMs.

For Answer-Only supervision, the trend is straightforward: larger policies consistently achieve higher Gold ELO, with steady improvements across scales. This aligns with the expectation that larger policies better exploit the reward signal and generalize more effectively.

For Thinking supervision, however, the picture is less clear. While the largest policy does achieve the highest peak performance, its Gold ELO curve saturates and bends downward earlier than smaller

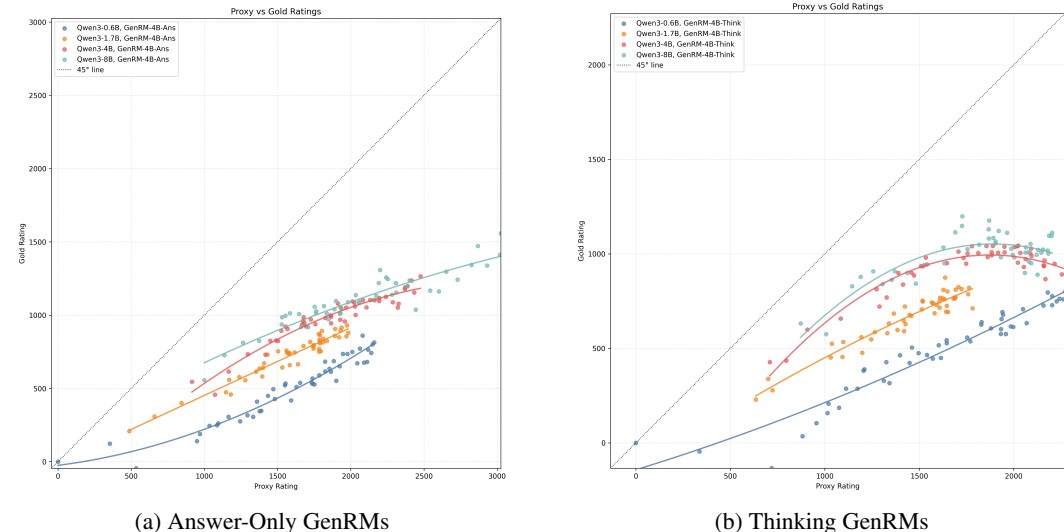

(a) Answer-Only GenRMs         (b) Thinking GenRMs

Figure 5: Proxy vs. Gold ELO ratings for policies of different sizes trained with fixed GenRMs. Left: Answer-Only GenRMs. Right: Thinking GenRMs. The dotted line indicates perfect alignment. **Takeaway:** Larger policies consistently improve under Answer-Only supervision, but under Thinking supervision, large policies saturate earlier, suggesting a more fragile interaction between policy and GenRM capacity.

policies, which continue to improve steadily. This suggests two possible explanations: (i) large policies may more quickly exhaust the effective capacity of the GenRM, reaching its "ceiling" earlier, or (ii) beyond a certain scale, further increasing policy size without correspondingly stronger GenRMs may become counterproductive.

Additional training is required to disentangle these explanations, but the evidence points toward an important asymmetry: scaling policy size is reliably beneficial under Answer-Only supervision, but under Thinking supervision, the interaction between policy capacity and GenRM capacity is more fragile.

### 4.6 EFFECT OF THE $\beta$ COEFFICIENT IN ONLINE DPO

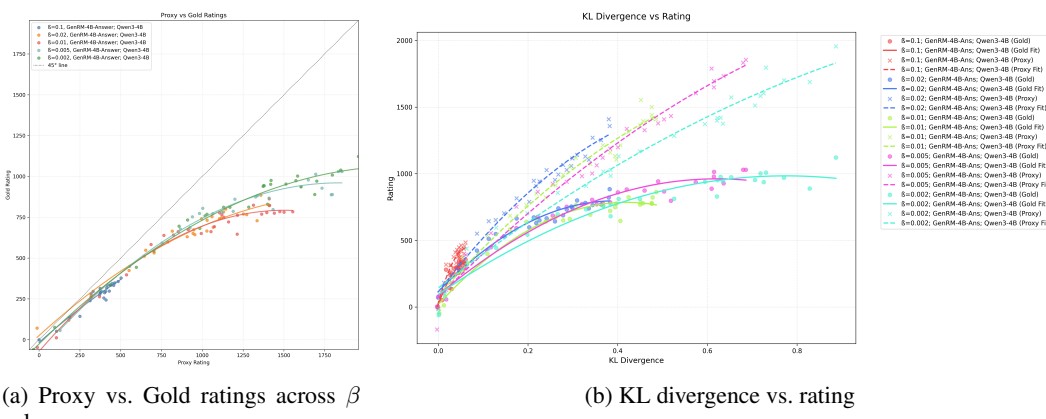

(a) Proxy vs. Gold ratings across $\beta$ values         (b) KL divergence vs. rating

Figure 6: Effect of the $\beta$ coefficient in Online DPO. Left: Proxy vs. Gold ELO ratings for Answer-Only GenRMs at different $\beta$ values. Right: KL divergence vs. rating tradeoff, showing how larger $\beta$ suppresses exploration. The dotted line indicates perfect alignment. **Takeaway:** Smaller $\beta$ values (0.002–0.01) yield higher Gold Elo by allowing more exploration, but the effect is secondary compared to GenRM or policy scaling. All policies have been trained on 50,000 prompts.

*How does the DPO regularization strength affect policy quality and the exploration-exploitation tradeoff?*

We analyze the effect of the $\beta$ coefficient in Online DPO. Figure 6(a) shows Proxy vs. Gold ELO ratings for policies trained with $\beta \in 0.1, 0.02, 0.01, 0.005, 0.002$ under Answer-Only supervision. Overall, smaller $\beta$ values consistently yield better Gold performance. While gains become increasingly incremental as $\beta$ decreases, the smallest value ($\beta = 0.002$) achieves the highest Gold ELO.

Importantly, lower $\beta$ does not simply increase Proxy ratings at the expense of Gold performance. Instead, the lowest $\beta$ exhibits the slope closest to the $45°$ line, indicating the least degree of over-optimization among all settings.

Figure 6(b) further illustrates the KL–performance tradeoff. Larger $\beta$ values overly constrain the policy to the reference distribution, limiting attainable Gold ratings. Reducing $\beta$ relaxes this constraint and improves alignment with Gold outcomes, without inducing severe divergence.

Overall, smaller $\beta$ values are strictly preferable in this regime. Although the marginal gains diminish at very low $\beta$, the smallest values simultaneously achieve the highest Gold ELO and the most faithful Proxy–Gold correspondence.

## 5 LIMITATIONS AND DISCUSSION

**Gold evaluator is a learned proxy, not humans.** We mitigate label noise by fine-tuning a 32B Gold evaluator on human preferences, re-annotating pairwise data with this model, and discarding items with inconsistent Gold verdicts (Gao et al., 2022). This improves consistency but couples our objective to the Gold model's inductive biases. At 32B parameters, the Gold judge is stronger than our proxies, leaving headroom for improvement, yet it still reflects preferences from its training set. *Takeaway.* Interpret results as optimization toward a strong, learned proxy.

**Answer-only format for the Gold evaluator.** The Gold judge emits a single verdict token. This choice improves throughput and simplifies adjudication, but might favor answer-only proxies in subtle ways. We investigated this concern empirically and found no evidence of systematic bias. First, off-the-shelf (untrained) answer-only and thinking judges show similar agreement patterns on the Gold-labeled set, with differences small relative to the trained-vs-baseline gap (Figure 1): at smaller sizes Thinking models slightly outperform, at mid-size they are comparable, and at larger sizes Answer-Only holds a small edge. Second, in Figure 2a (baseline proxies), policies trained with thinking baseline judges trace a slope closer to the diagonal, indicating tighter proxy–Gold agreement even though Gold is answer-only. These observations suggest that the Answer-Only advantage we report in trained GenRMs reflects genuine optimization dynamics rather than evaluator format bias. Nonetheless, we do not evaluate an alternative thinking-style Gold, which would further strengthen robustness. *Takeaway.* Empirical checks show no format bias at baseline; a rationale-producing Gold remains valuable future work.

**Domain scope.** All experiments use creative writing; we do not systematically evaluate other domains such as safety, helpfulness, or verifiable tasks (math, code). Replicating our full experimental grid in additional domains would scale cost roughly linearly: each (judge, policy) configuration requires an online DPO run plus Elo evaluation, costing 6–20 hours on $8\times$H100, not including judge training. Given our ablation count (policy size $\times$ judge size $\times$ judge mode $\times$ training-budget/$\beta$/checkpoint sweeps), full multi-domain replication would require thousands of additional H100-hours. To provide preliminary evidence, we ran cross-domain checks on OpenOrca using off-the-shelf Qwen3 models (Appendix A.1); these support the same qualitative pre-training trend (thinking judges start stronger as static evaluators). *Takeaway.* Our conclusions are scoped to non-verifiable creative writing; domain transfer is an important open question.

**Relation to prior creative-writing studies.** Creative-writing preferences are subjective. Optimizing win rate can reduce stylistic diversity. We observe some style convergence (Chung et al., 2025) in late-stage policies against both judge modes, while draw rates do not show large collapses. Fein et al. (2025) report that chain-of-thought can degrade verification accuracy for creative writing, and that trained BT and generative verifiers outperform zero-shot judges on their benchmark. In our setting, distilling Qwen3-32B traces into Qwen3-4B also yields negligible gains as evaluators, which aligns with these observations. However, after training, thinking judges significantly improve as

static evaluators. The gap between thinking and answer-only during policy optimization therefore cannot be explained by domain "unsuitability" alone; it indicates different optimization dynamics. *Takeaway.* Our evaluation–optimization divergence is a property of the training loop in this domain, not only a property of static judging.

**Family and algorithm scope, and behavioral priors.** All models use Qwen3 backbones where pretraining data are not public. Cross-family studies suggest that behavioral priors, including synthetic data that instantiate verification or backtracking, can modulate RL improvements and collapse family gaps (Gandhi et al., 2025). Such priors could shift our coefficients. We do not evaluate PPO-style RLHF for policies or alternative thinking-judge recipes. Additionally, we do not include Best-of-N (BoN) inference-time comparisons; our focus is on GenRMs as *training signals* for policy optimization rather than inference-time scaling, where the primary gains in our setting arise from gradient-based learning rather than reranking at generation time. BoN comparisons remain valuable future work for understanding the full compute-performance tradeoff. *Takeaway.* Our coefficients and inflection points are conditional on online DPO and Qwen3 underlying behavioral priors.

**Elo anchoring and schedule.** Elo is anchored to the Gold evaluator and depends on the match schedule (Chiang et al., 2024). We report both global and size-stratified arenas, but we do not study alternative anchors or tournament designs. *Takeaway.* Absolute Elo levels can shift with the anchor, while within-arena orderings are more stable.

## 6 RELATED WORK

Alignment from preferences exhibits predictable overoptimization: gold reward degrades as policies drift from a reference, following smooth scaling laws for both RLHF (Ouyang et al., 2022; Gao et al., 2022) and direct methods like DPO that remove explicit reward heads (Rafailov et al., 2023; 2024). This motivated architecturally unified judges. Generative Reward Models (GenRMs) replace scalar heads with next-token prediction, enabling rationales alongside verdicts (Mahan et al., 2024; Zhang et al., 2025). Mahan et al. (2024) trained GenRMs via iterative self-taught reasoning with DPO, achieving strong out-of-distribution generalization. Subsequent work scales these approaches: J1 extends with GRPO and positional consistency rewards (Whitehouse et al., 2025), DeepSeek-GRM adds Self-Principled Critique Tuning with meta-aggregation for inference-time scaling (Liu et al., 2025), and Heimdall demonstrates test-time improvements via majority voting in verification tasks (Wang et al., 2025; Shi & Jin, 2025). Complementary supervision strategies include self-generated critiques (Yu et al., 2025) and criteria trees (Liang et al., 2025), while EvalPlanner frames evaluation as plan-and-reason generation (Saha et al., 2025). Despite extensive work on judge reliability (Ye et al., 2024) and benchmarks (Lambert et al., 2024), the field conflates static evaluation accuracy with rewarder effectiveness. We disambiguate these roles through controlled scaling experiments with answer-only (SFT) versus thinking (GRPO) GenRMs as both evaluators and online DPO rewarders, using Elo arenas (Chiang et al., 2024) for unified comparison. Our results reveal when inference-time reasoning helps evaluation but hinders policy optimization under matched FLOPs and KL budgets.

## 7 CONCLUSION

We present the first systematic study of generative reward models as online training signals. Our experiments reveal a consistent *evaluator-rewarder gap*: Thinking GenRMs outperform Answer-Only variants on static evaluation, yet this advantage reverses during policy optimization, where Answer-Only models yield higher Gold Elo and more stable alignment.

Three findings offer practical guidance: (1) reward model scale dominates other factors, with gains continuing even when the GenRM far exceeds the policy in size; (2) intermediate GenRM checkpoints can outperform fully-trained models as rewarders; and (3) in-distribution accuracy does not predict training effectiveness. These results suggest that optimizing GenRMs for static benchmarks may be counterproductive for downstream alignment.

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

# A  APPENDIX

Additional experimental details and supplementary results.

## A.1  PRELIMINARY CROSS-DOMAIN EVALUATION: OPENORCA

To provide preliminary evidence on domain generalization (see §5), we ran cross-domain checks on OpenOrca using off-the-shelf Qwen3 policies, judges, and Gold evaluator. No fine-tuning was performed. These results support the same qualitative pre-training trend observed in the main creative-writing domain: prior to GenRM training, Thinking judges show stronger alignment with the Gold evaluator than Answer-Only judges.

## A.2  ELO RATING COMPUTATION

Given a set of policy checkpoints $\mathcal{S}$, we generate pairwise matches $(i, j) \in \mathcal{S} \times \mathcal{S}$ on held-out prompts and obtain Gold evaluator decisions $w_{ij} \in \{0, 1\}$. Elo ratings $\{R_s\}_{s \in \mathcal{S}}$ are estimated by maximizing the logistic likelihood:

$$\max_{\{R_s\}} \sum_{(i,j)} \left[ w_{ij} \log \sigma\left( \frac{R_i - R_j}{s} \right) + (1 - w_{ij}) \log \sigma\left( \frac{R_j - R_i}{s} \right) \right],$$

with scale $s$ fixed. Proxy Elo ratings are computed analogously using proxy GenRM decisions in place of Gold decisions.

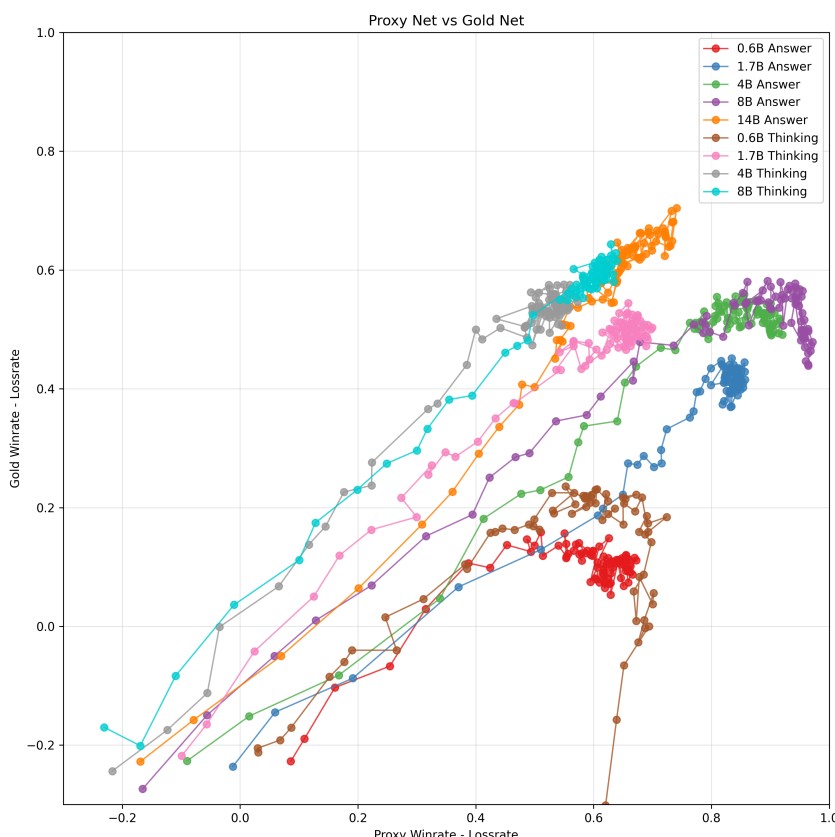

Figure 7: Proxy vs. Gold net score on OpenOrca. All models are off-the-shelf (no GenRM training). A full replication with trained GenRMs and policy optimization remains future work.

## B  PROMPT TEMPLATES

We provide the exact prompt templates used for policy generation and GenRM pairwise evaluation. All models receive these prompts without modification.

### B.1  POLICY PROMPT

The policy model receives the following prompt template for creative writing generation:

```
Creative Writing Prompt:
{QUESTION}
Goal: write an original short story that directly answers the prompt.
Requirements: Output only the story text. no titles, notes, commentary
or meta-text.
```

where {QUESTION} is replaced with the specific creative writing instruction from the evaluation set.

### B.2  GENRM PAIRWISE EVALUATION PROMPT

The generative reward model receives the following prompt template for pairwise comparison:

```
Two AI models were given the same instruction and each produced a reply.
Your task is to judge which reply better fulfills the instruction.
```

```
[BEGIN INSTRUCTIONS GIVEN TO BOTH MODELS]
{POLICY_PROMPT}
[END INSTRUCTIONS]

[BEGIN REPLY A]
{RESPONSE_A}
[END REPLY A]

[BEGIN REPLY B]
{RESPONSE_B}
[END REPLY B]

Output ONLY one of:
<answer>[[A]]</answer>
<answer>[[B]]</answer>
```

where {POLICY_PROMPT} contains the full policy prompt with the specific question, and {RESPONSE_A} and {RESPONSE_B} contain the two candidate responses to be compared. The model outputs a structured verdict indicating which response better satisfies the original instruction.

# C SUPPLEMENTARY FIGURES

## C.1 BETA COEFFICIENTS

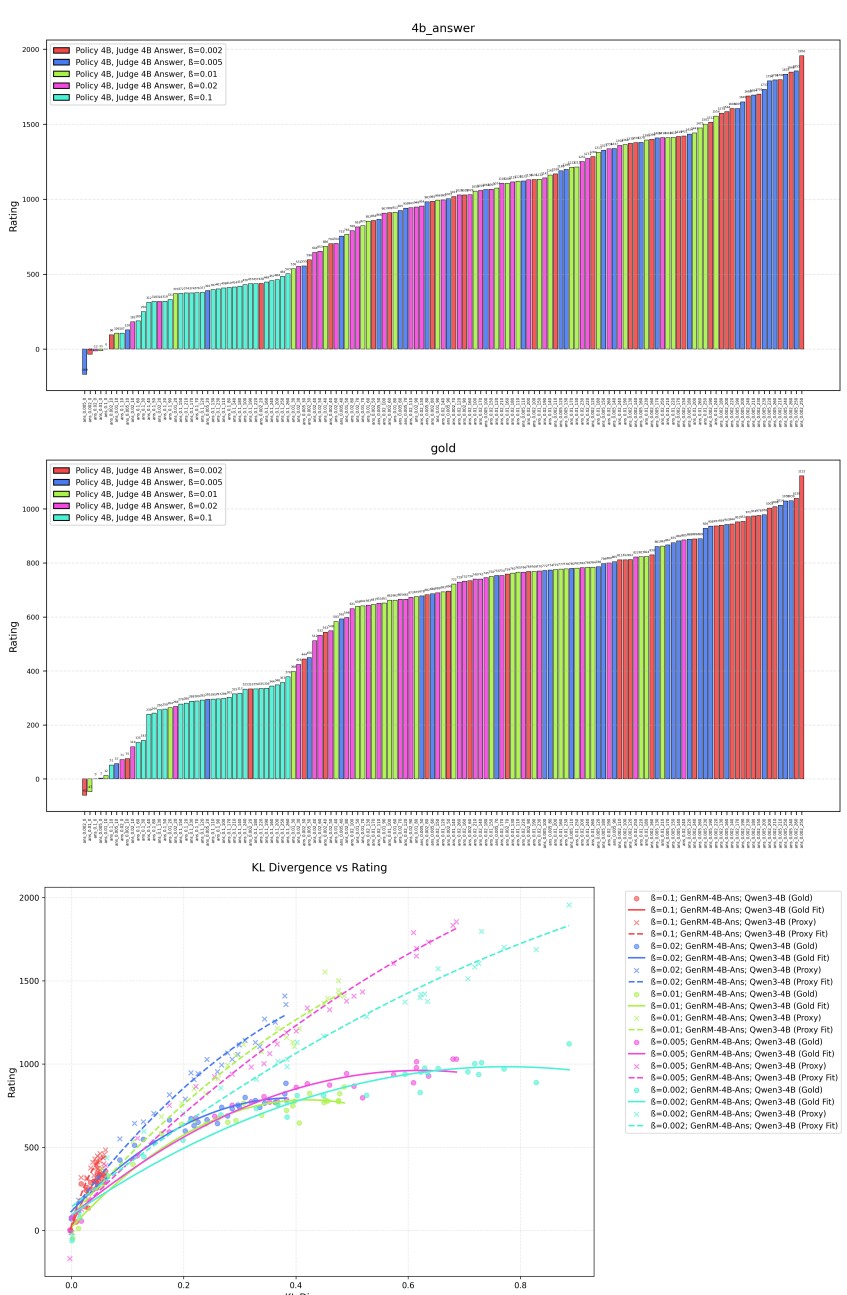

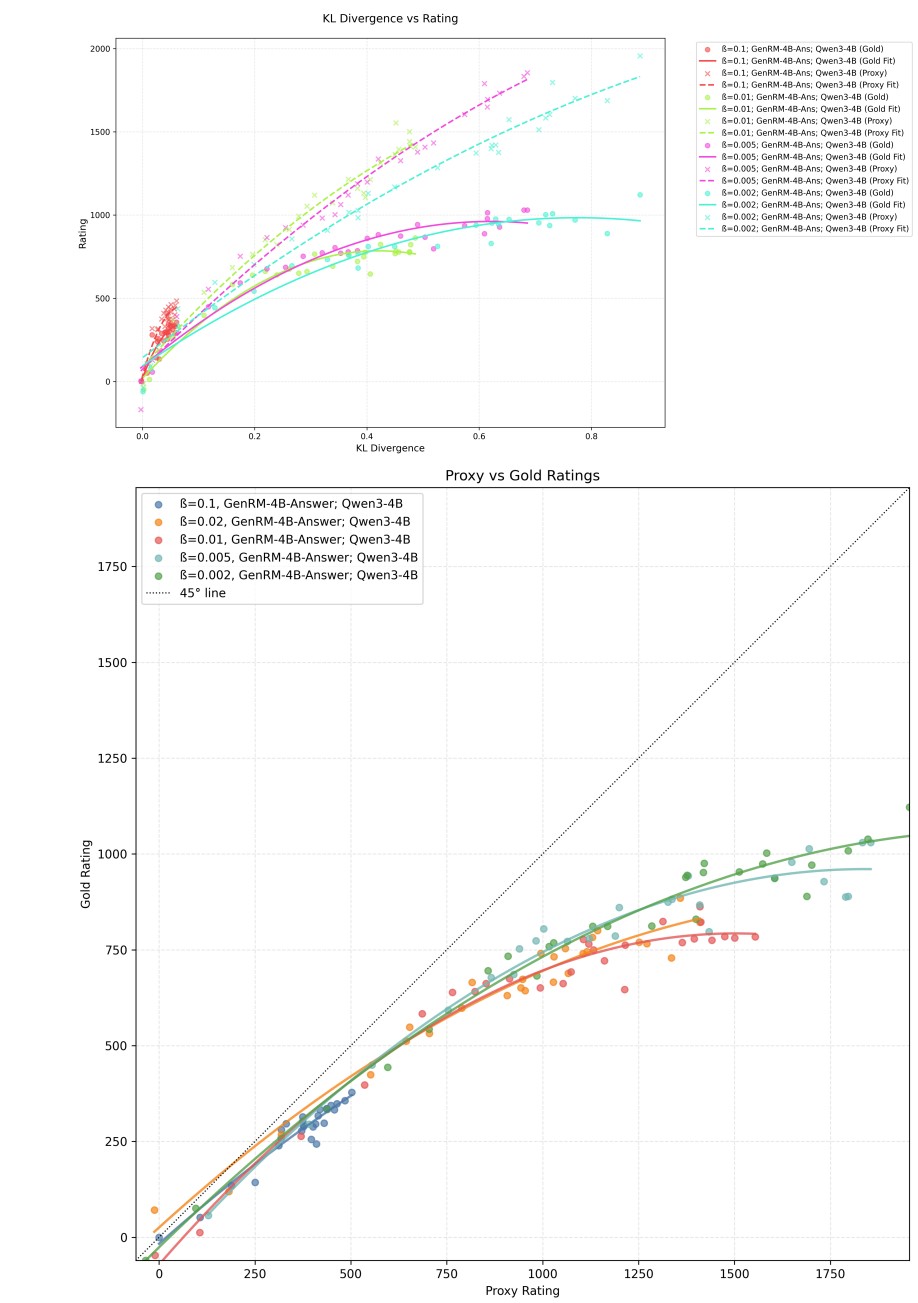

864
865
866
867
868
869
870
871
872
873
874
875
876
877
878
879
880
881
882
883
884
885
886
887
888
889
890
891
892
893
894
895
896
897
898
899
900
901
902
903
904
905
906
907
908
909
910
911
912
913
914
915
916
917

## C.2   GENRM EVALUATION

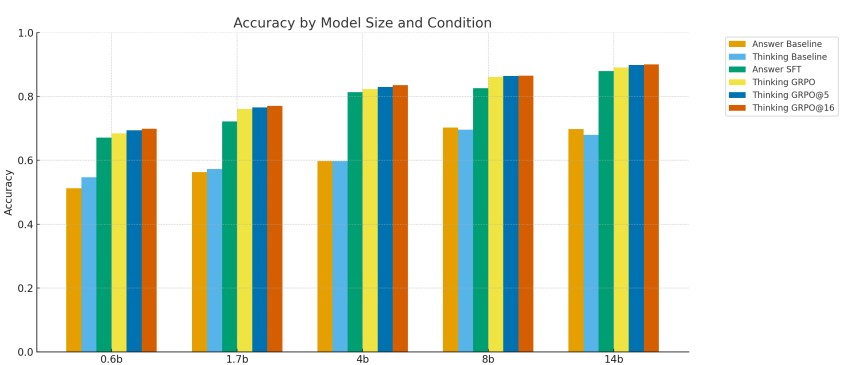

## C.3   GENRM SIZES (ANSWER)

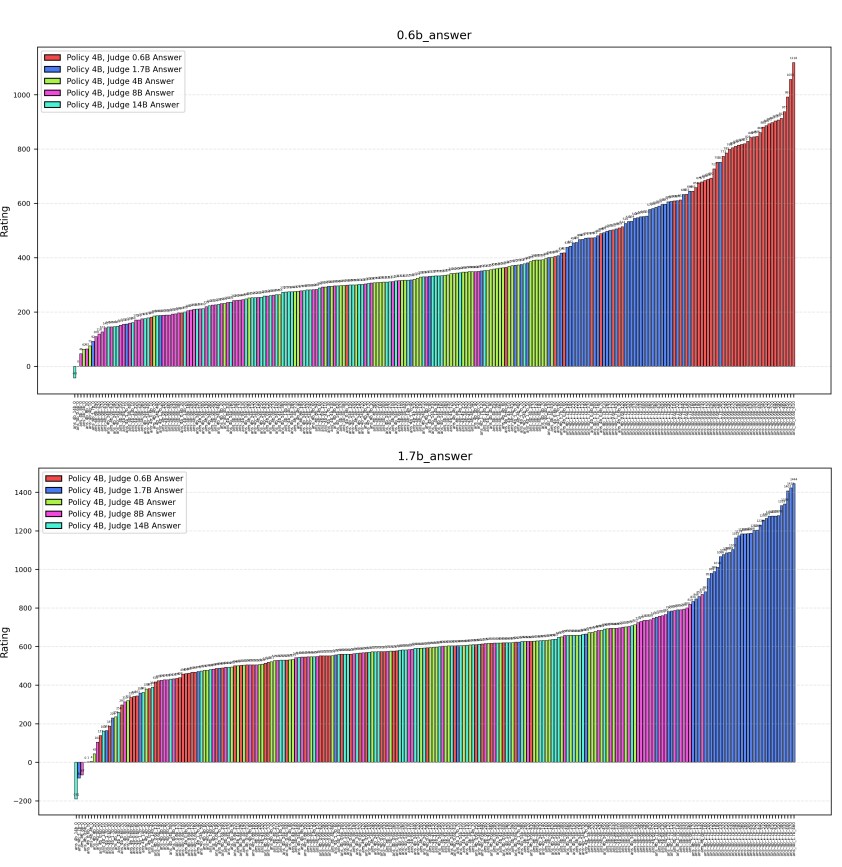

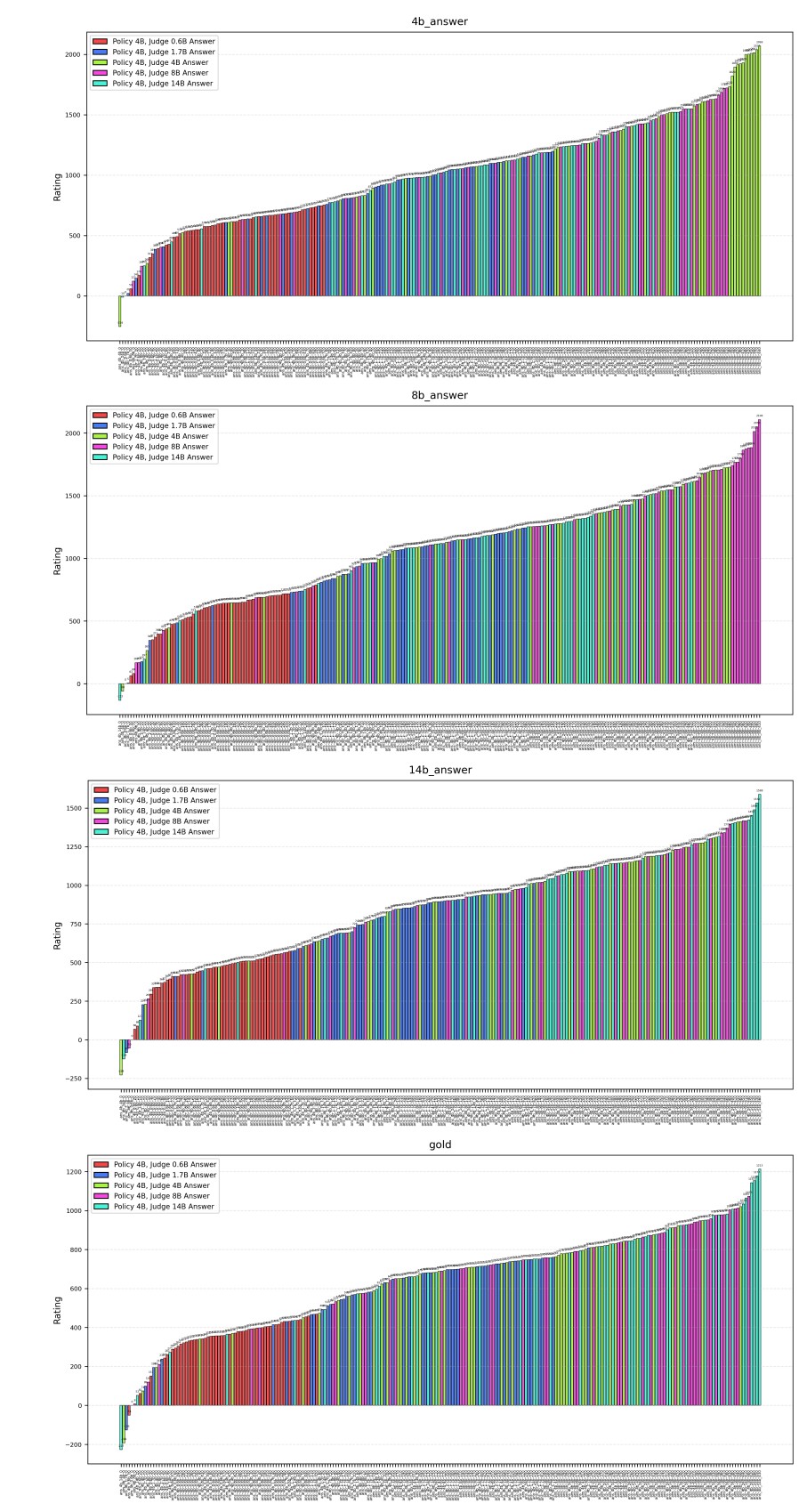

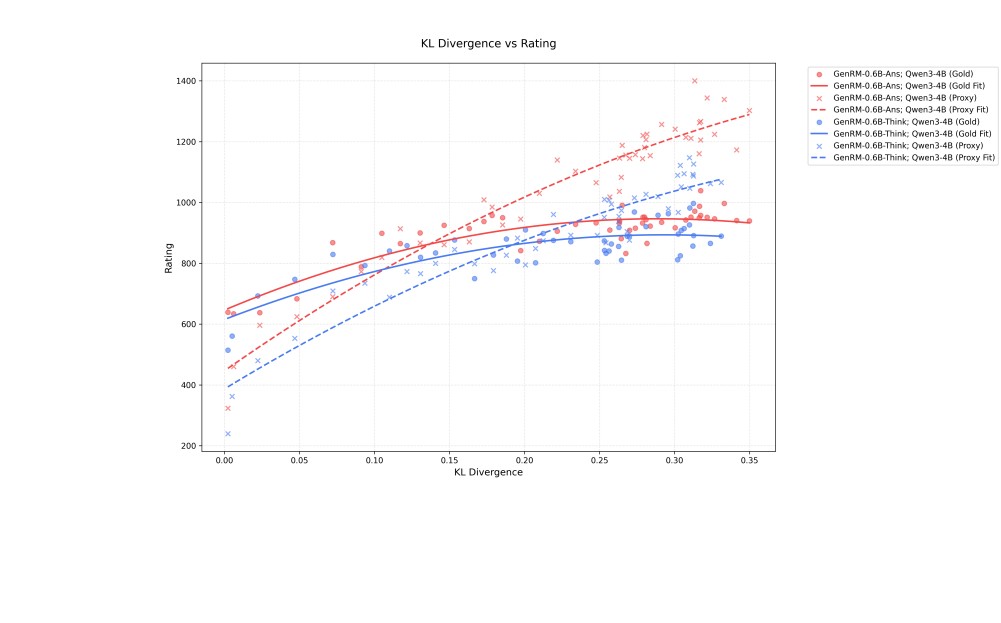

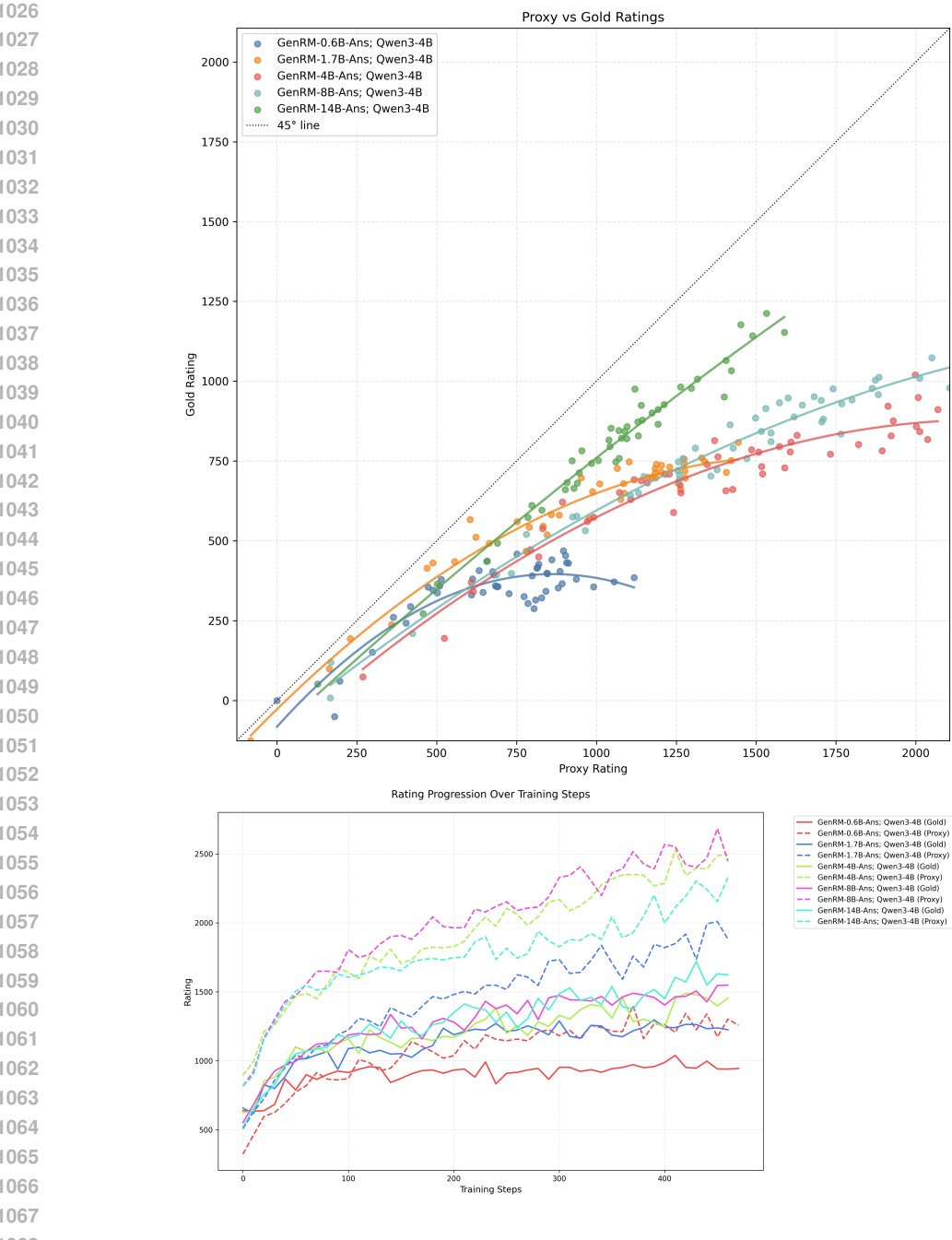

## C.4 GENRM SIZES (THINKING)

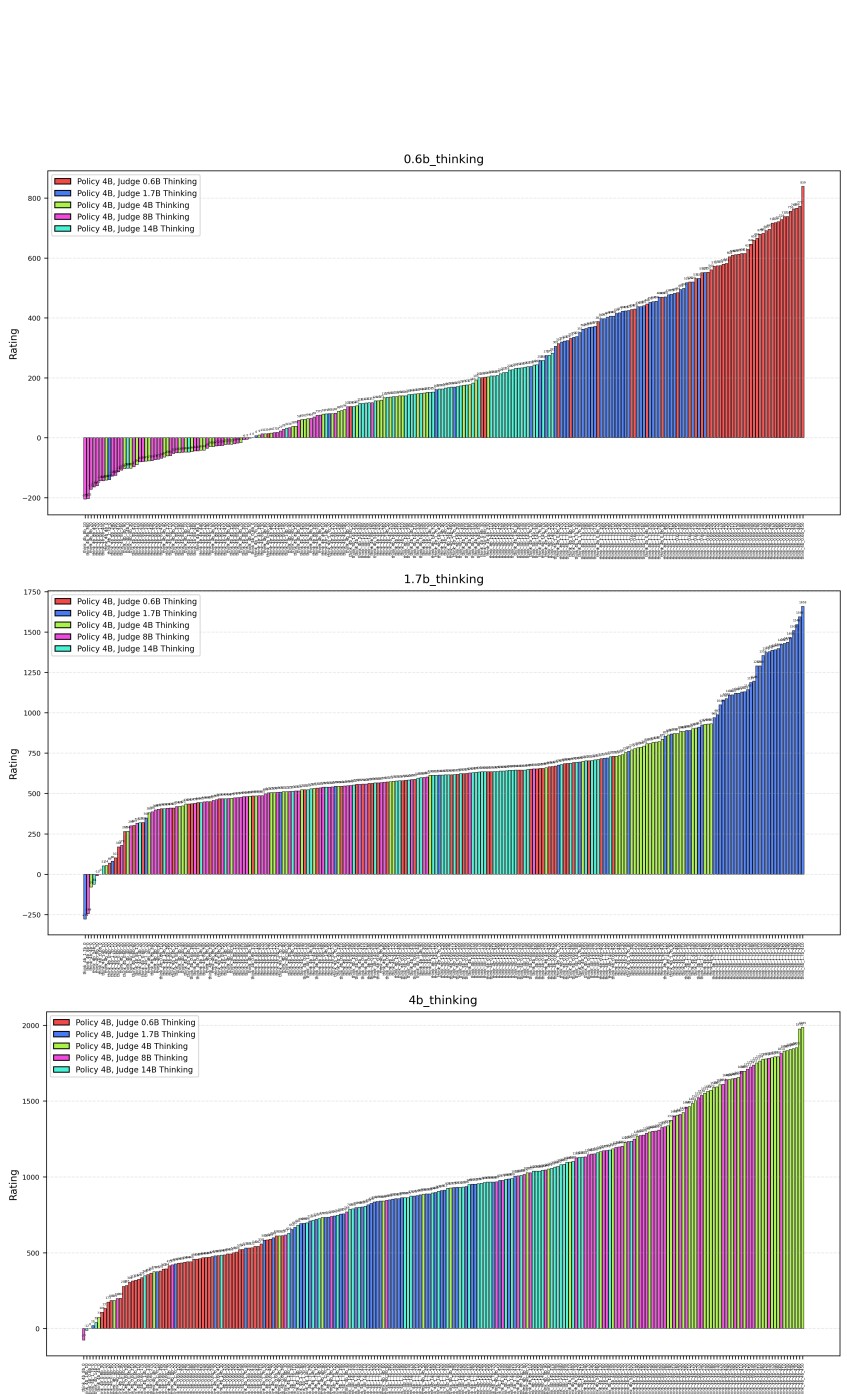

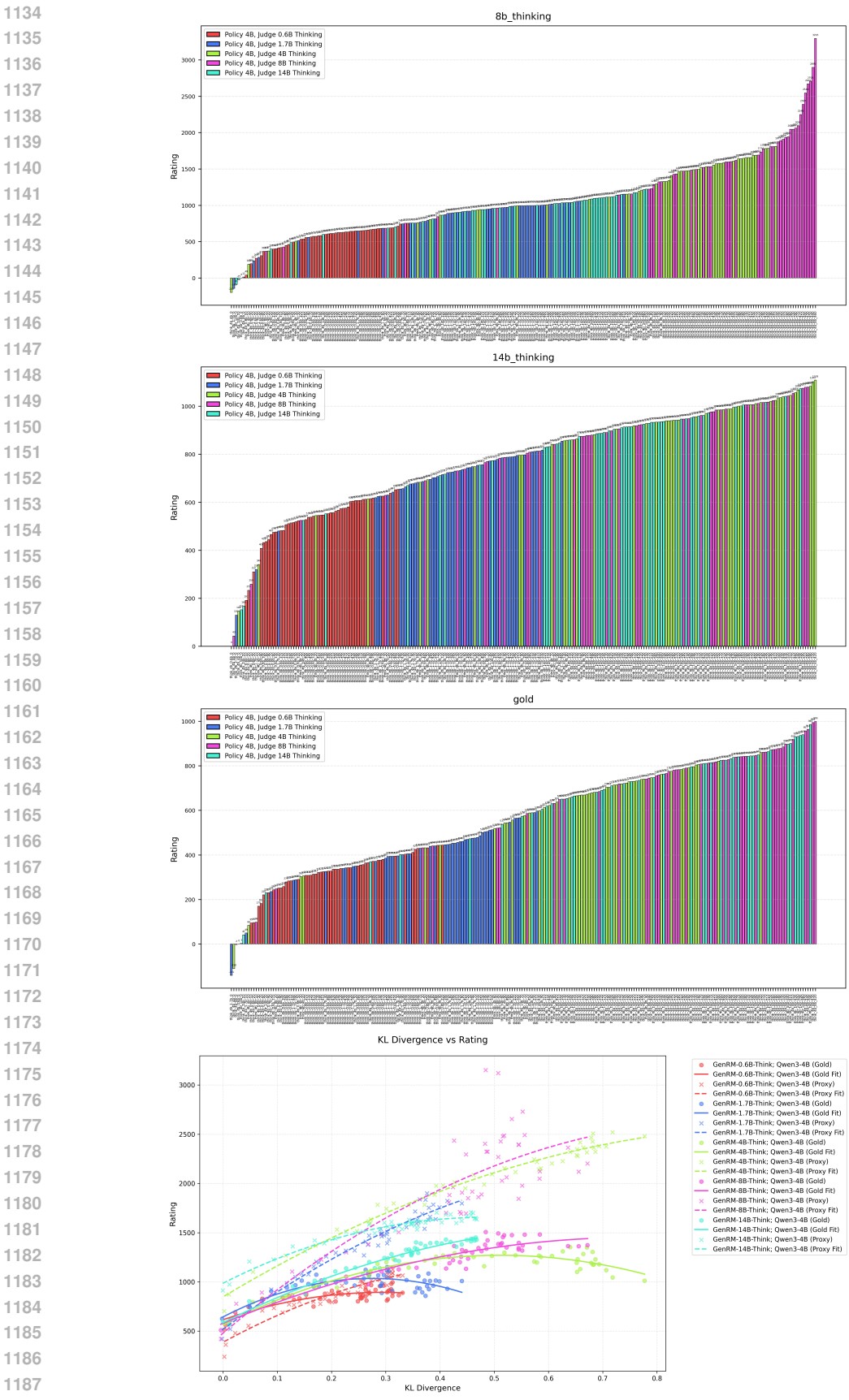

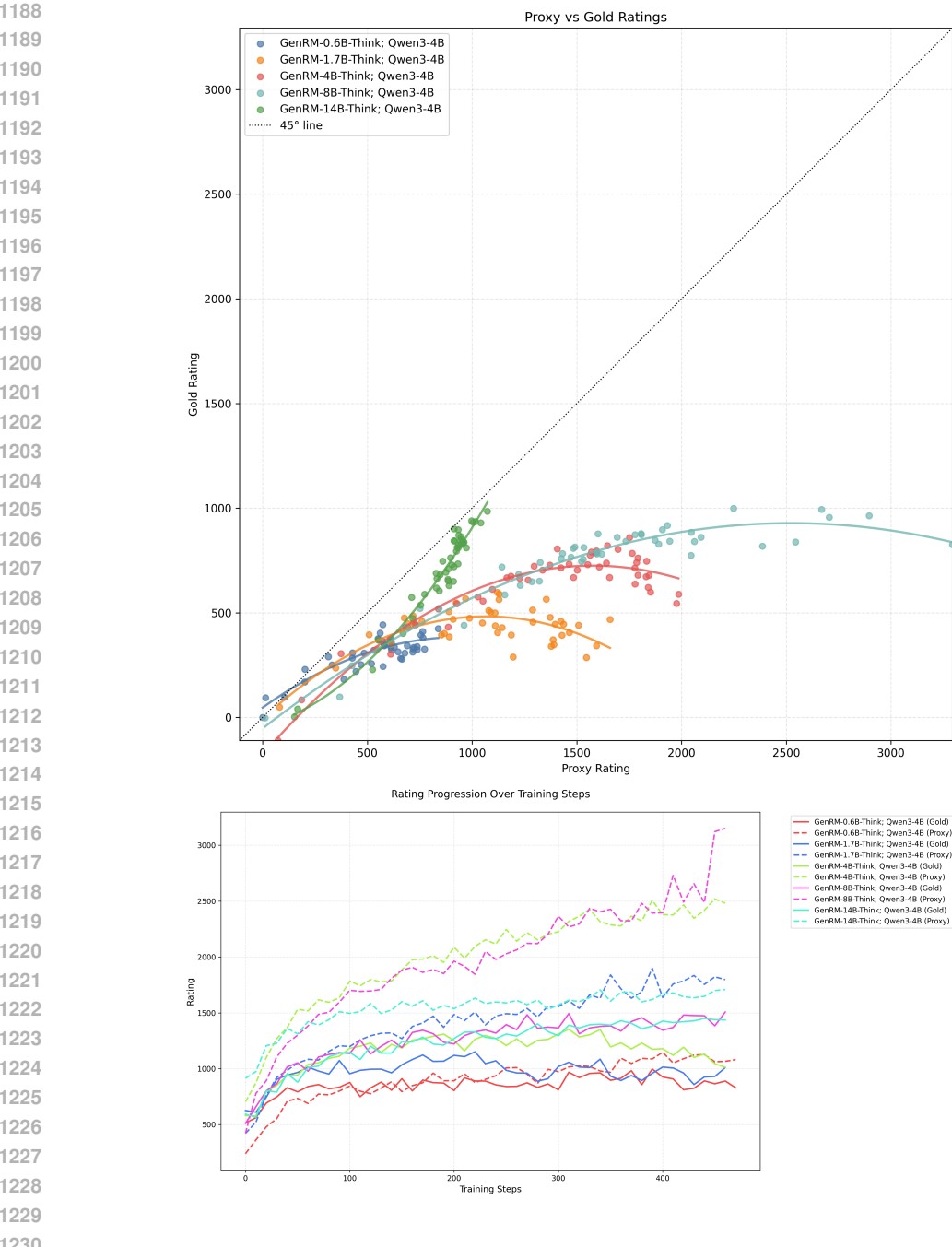

## C.5    PARAM BALANCE (ANSWER)

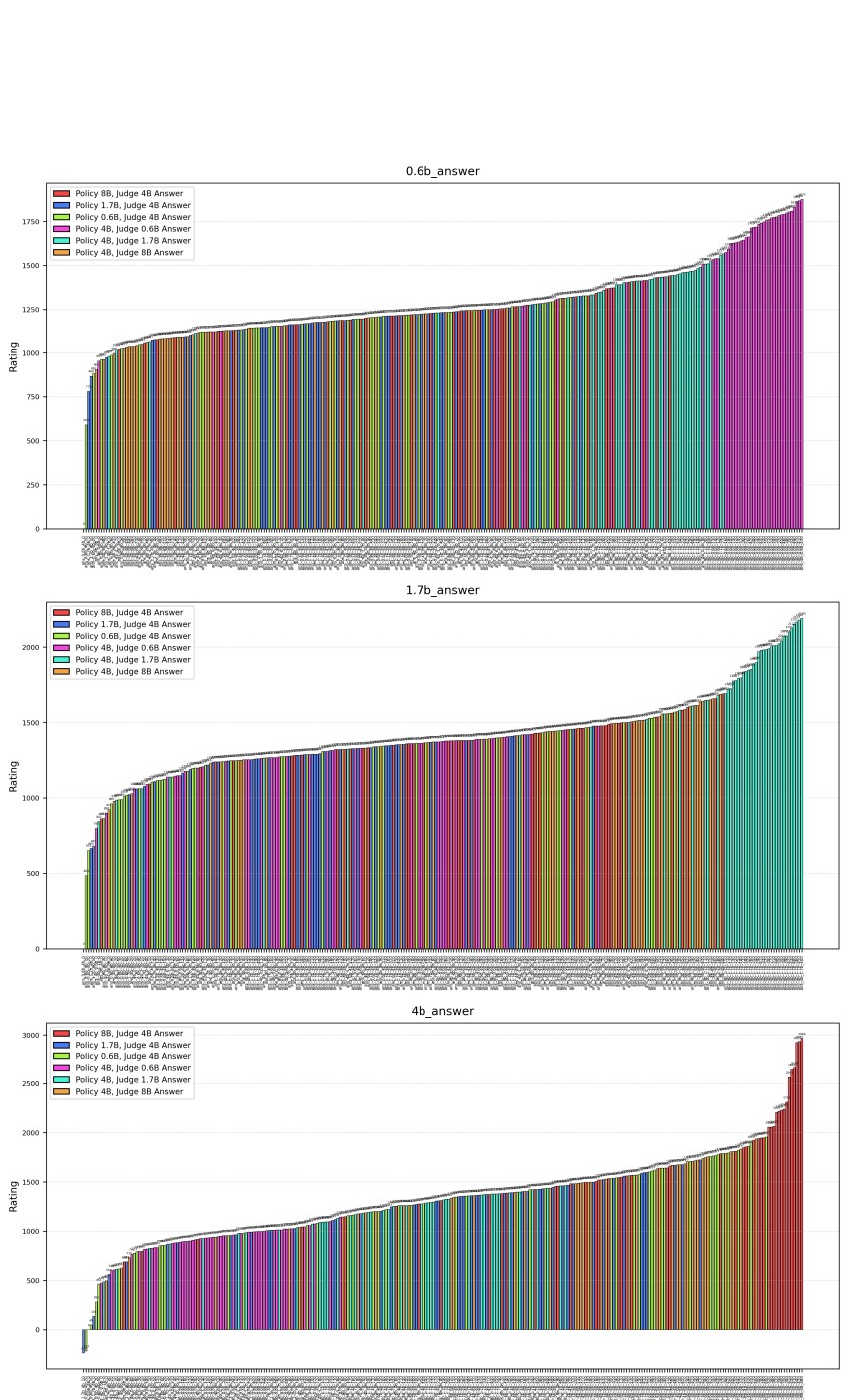

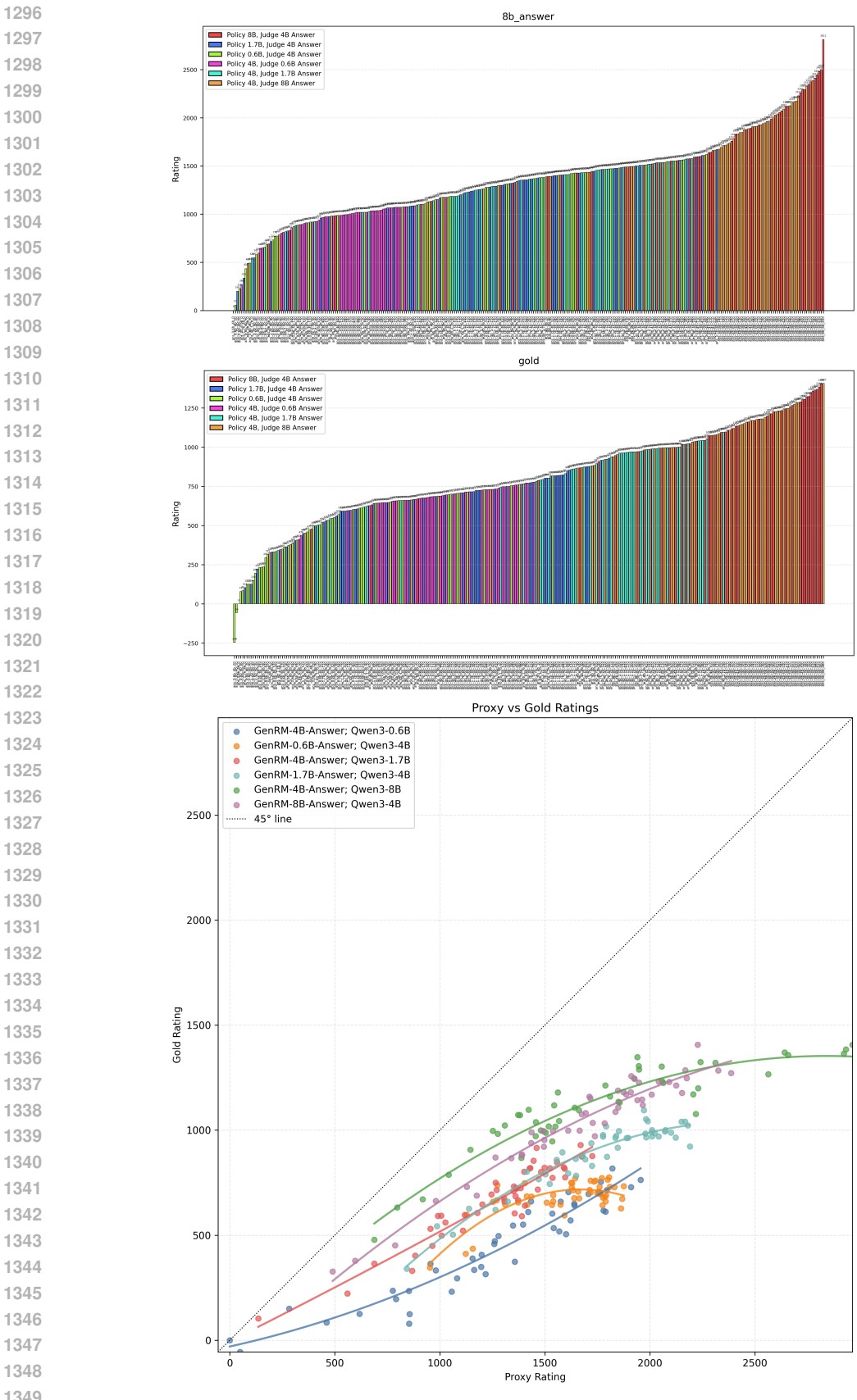

## C.6 PARAM BALANCE (THINKING)

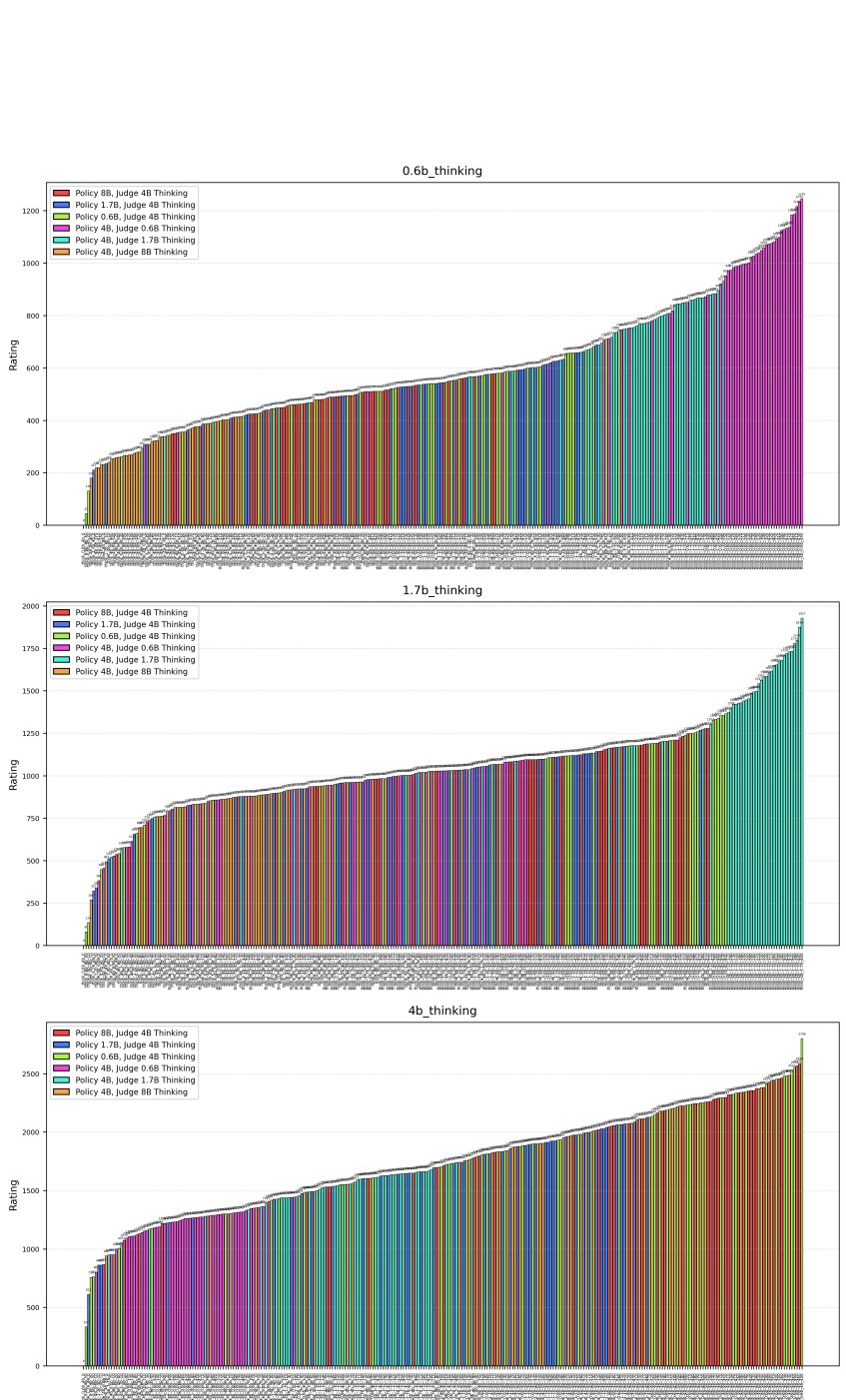

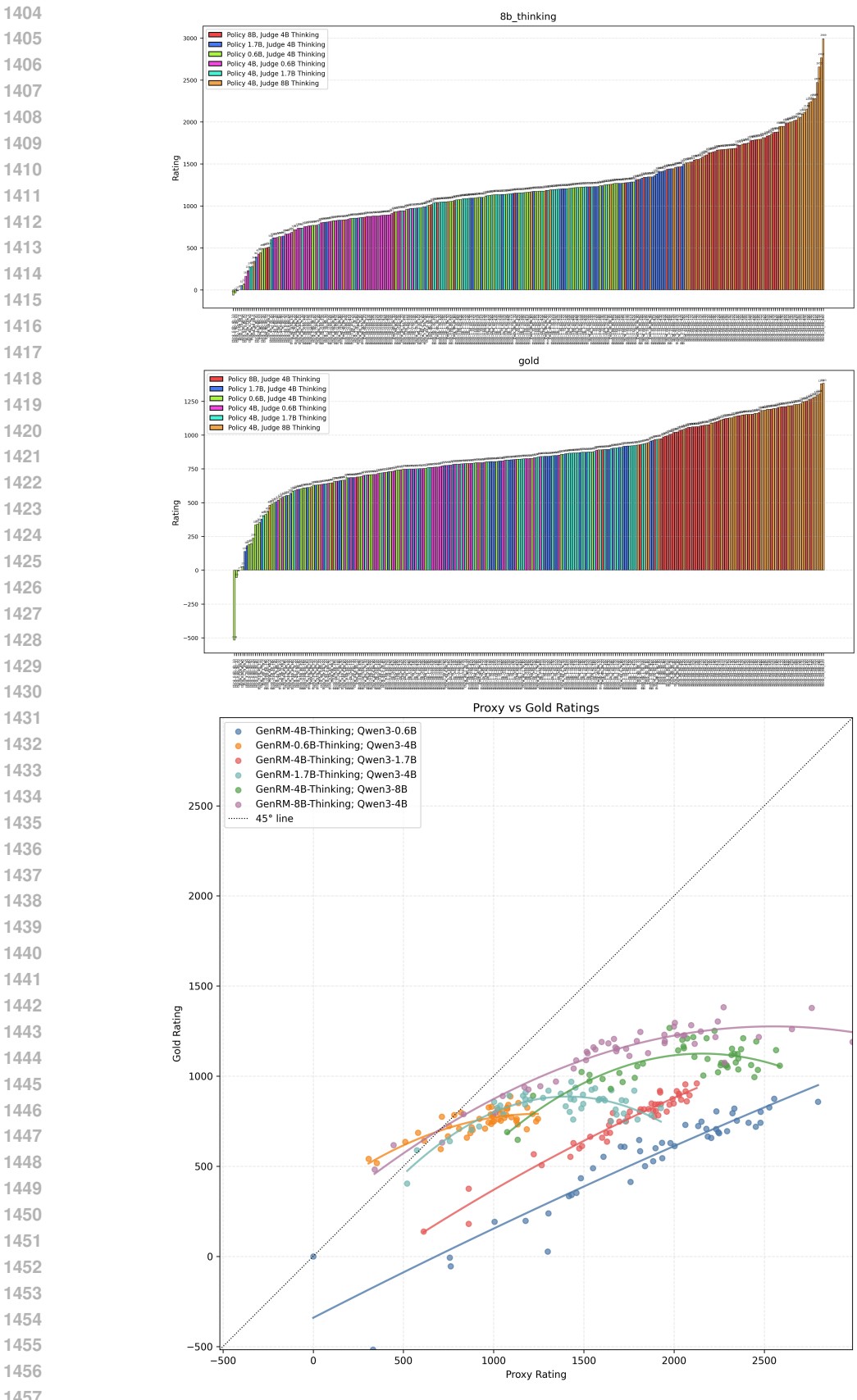

## C.7 POLICY SIZES (ANSWER)

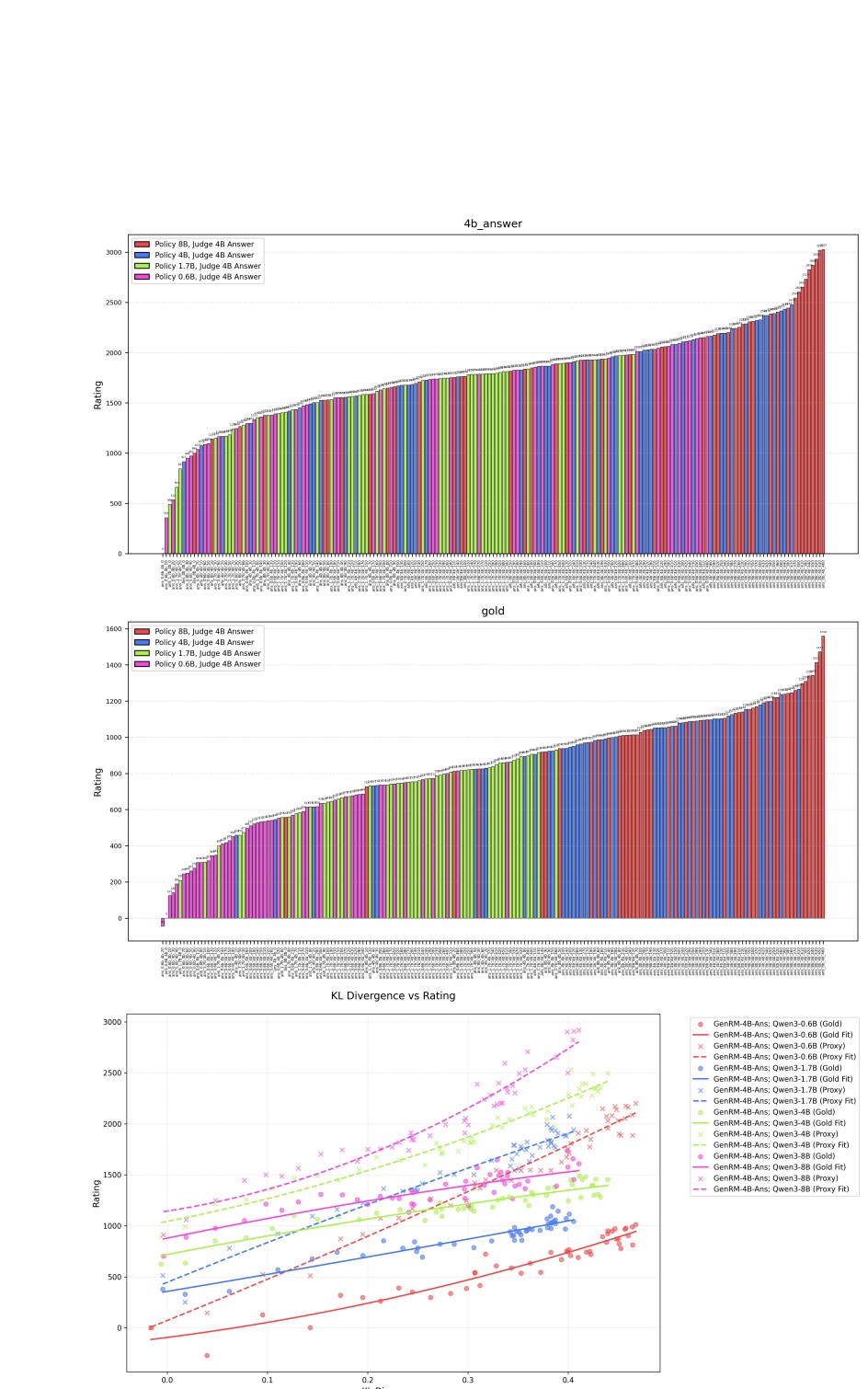

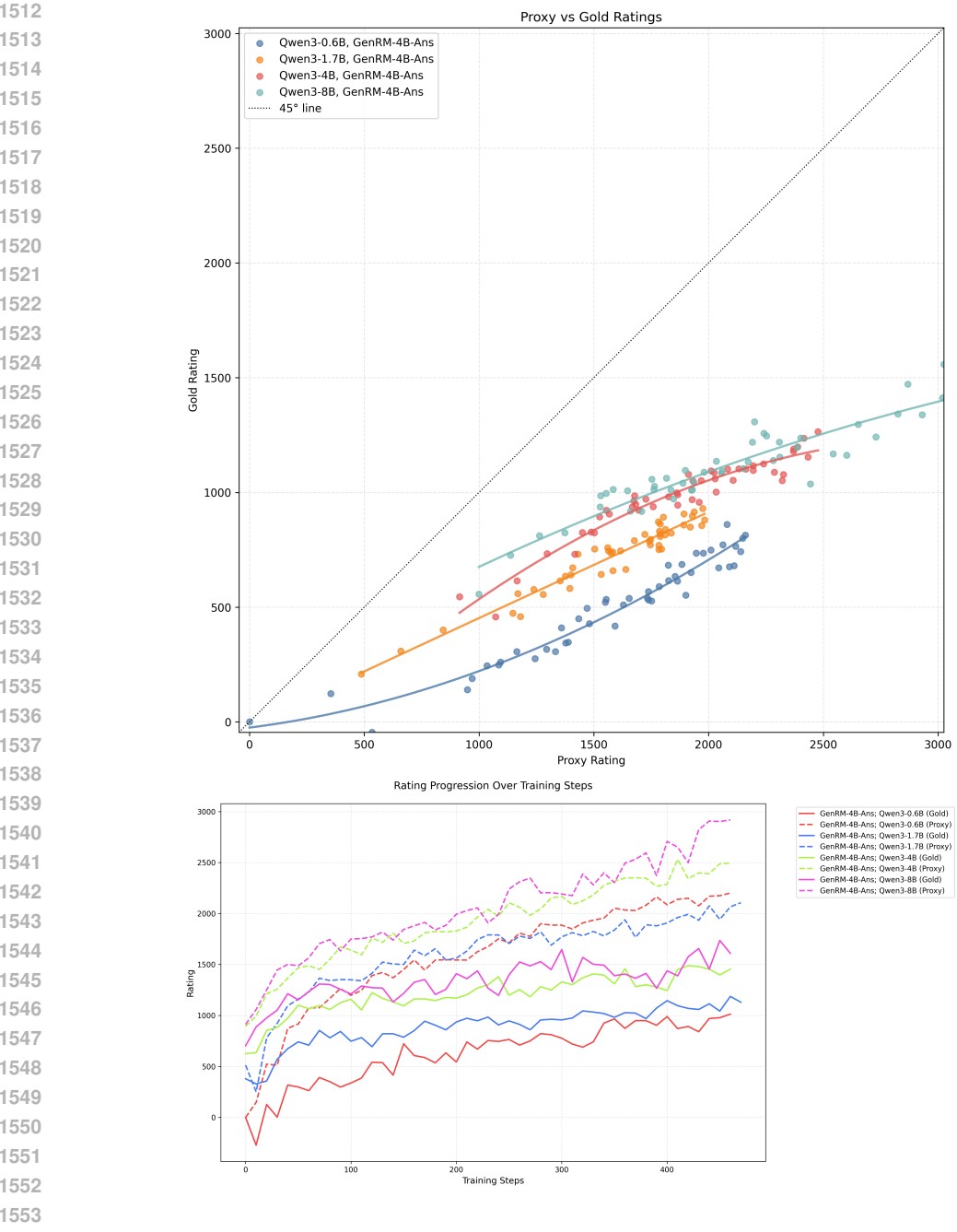

## C.8 POLICY SIZES (THINKING)

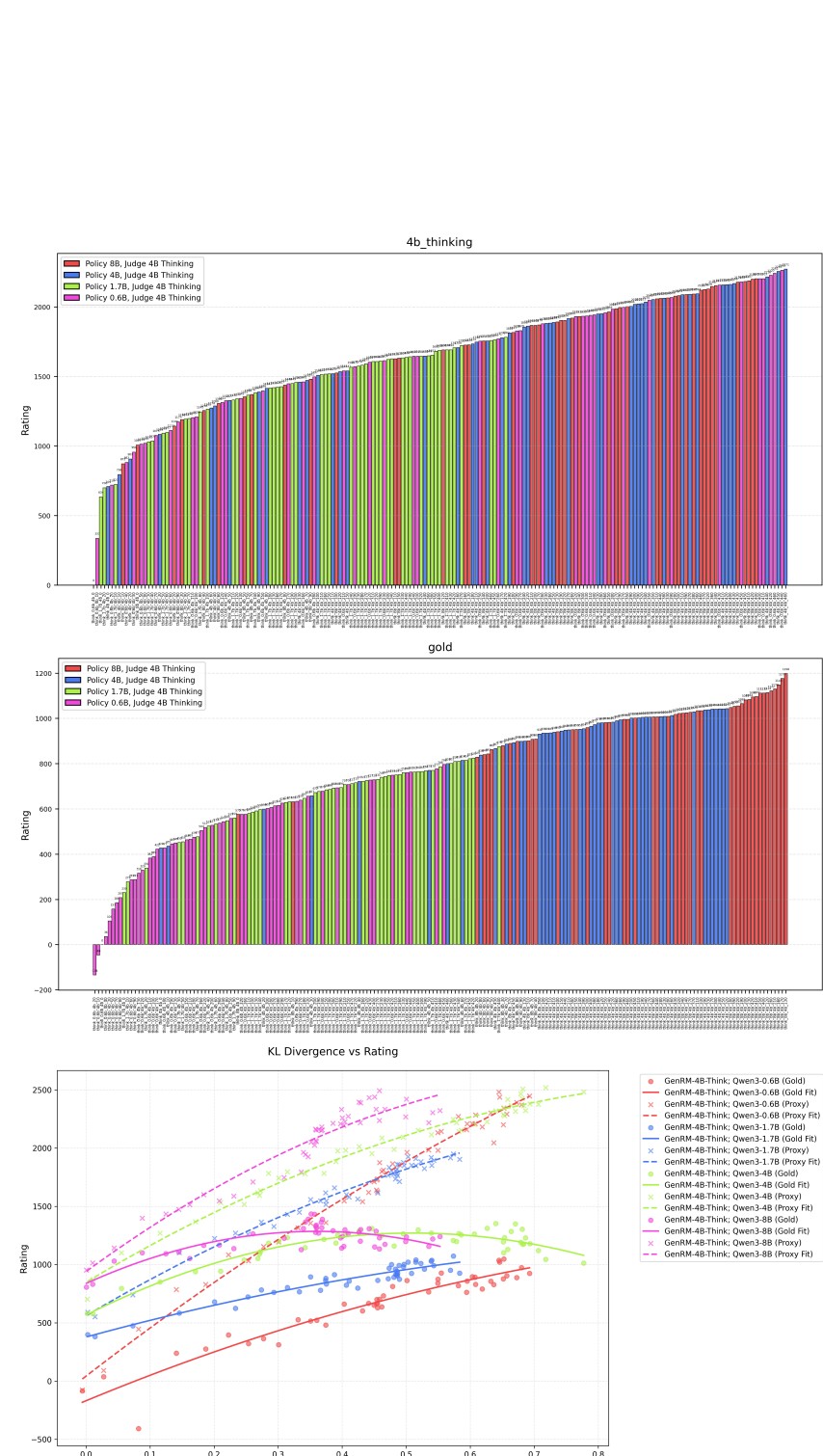

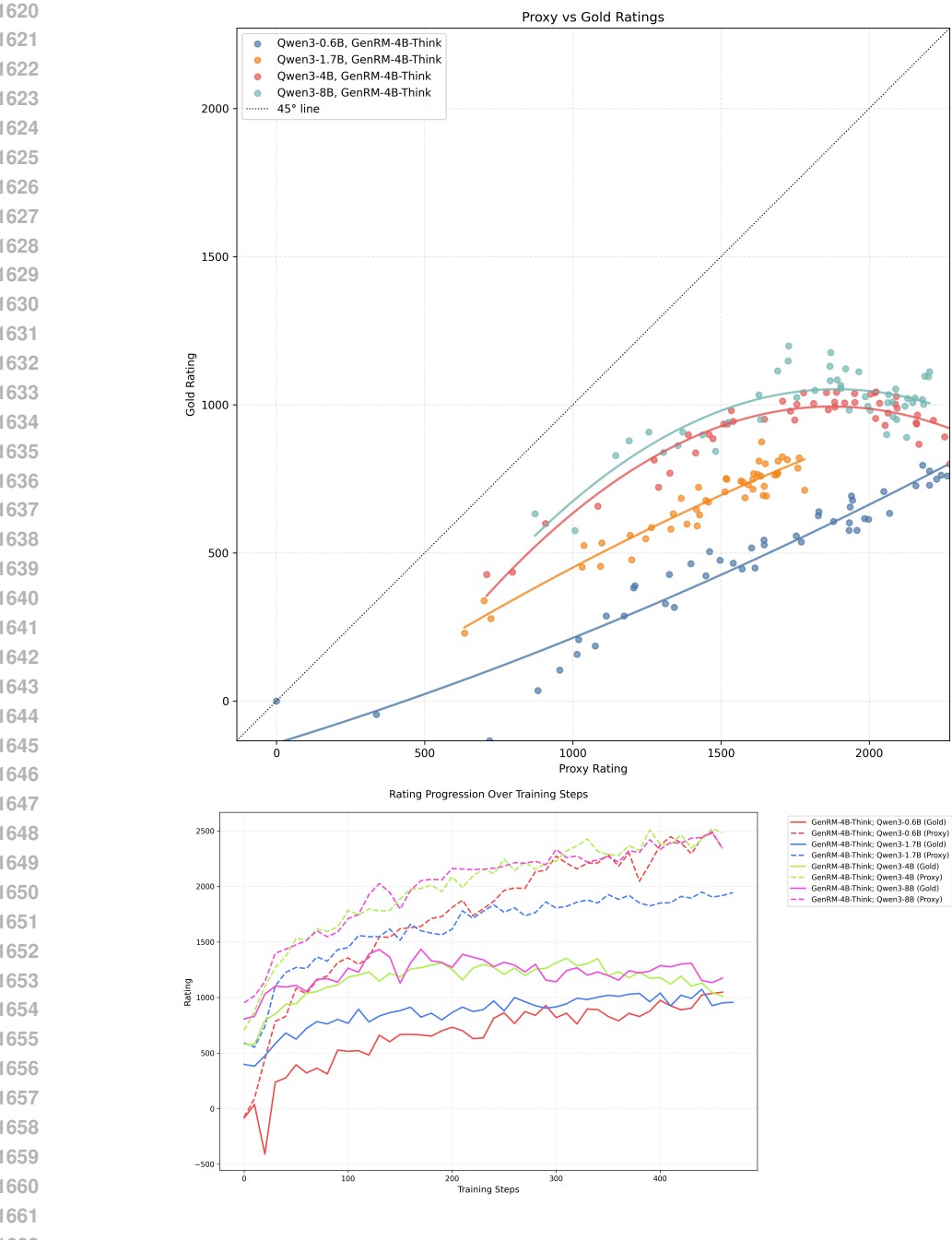

C.9  THINKING VS ANSWER (0.6B VS 4B)

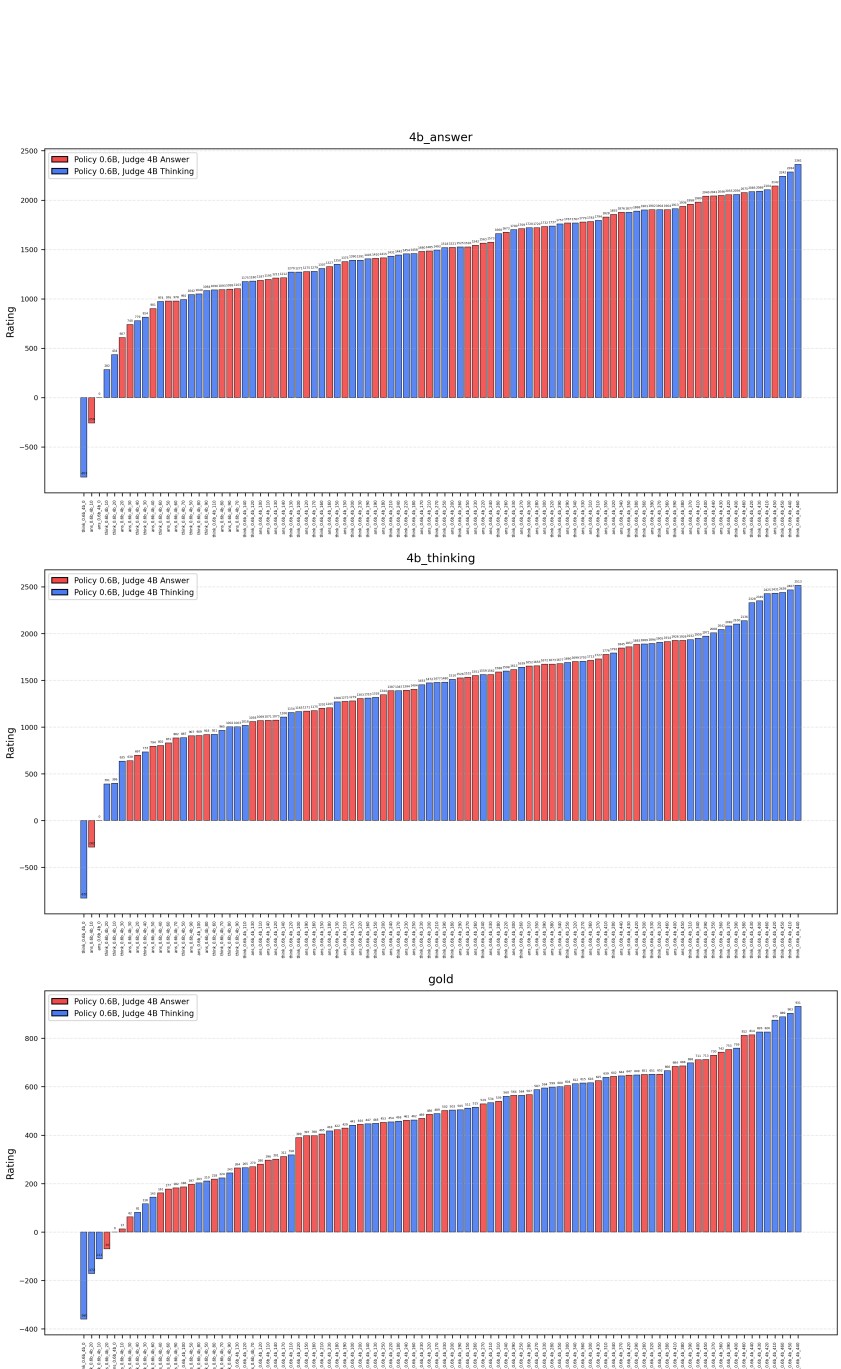

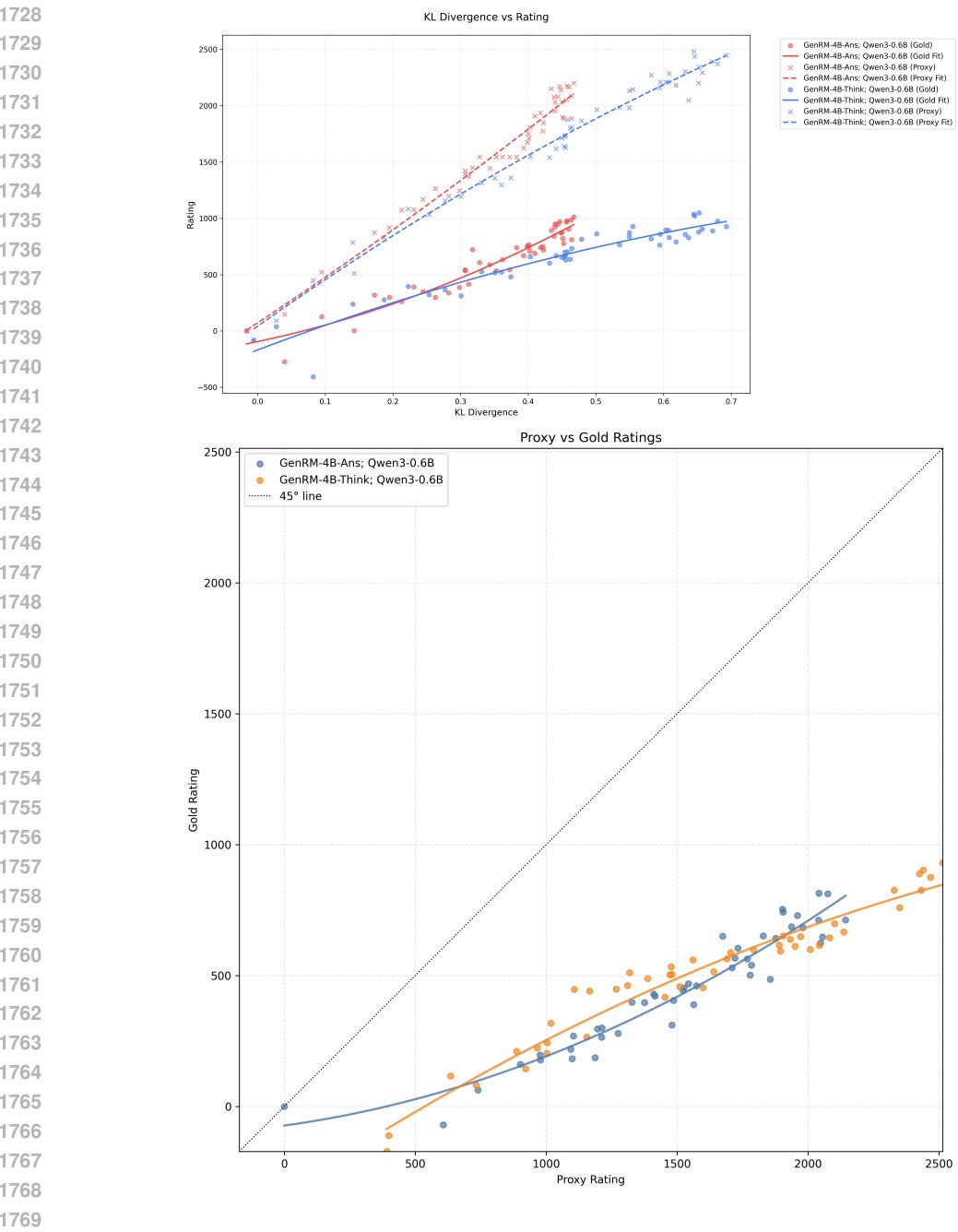

## C.10 THINKING VS ANSWER (1.7B VS 4B)

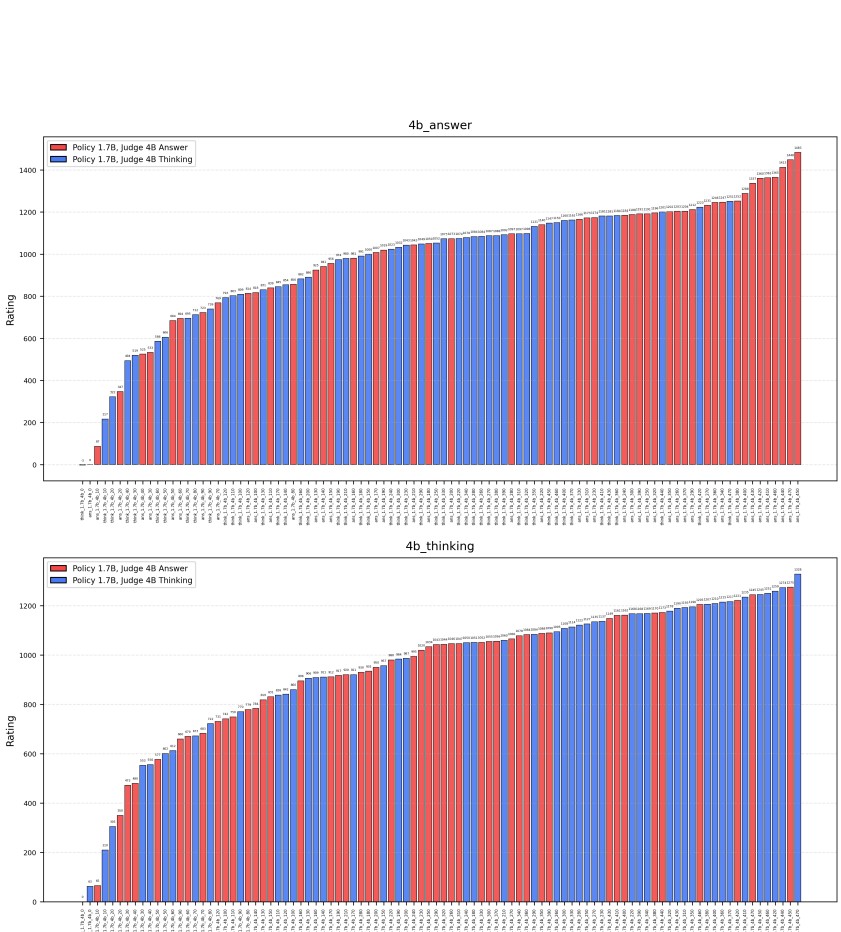

## C.11 THINKING VS ANSWER (4B VS 0.6B)

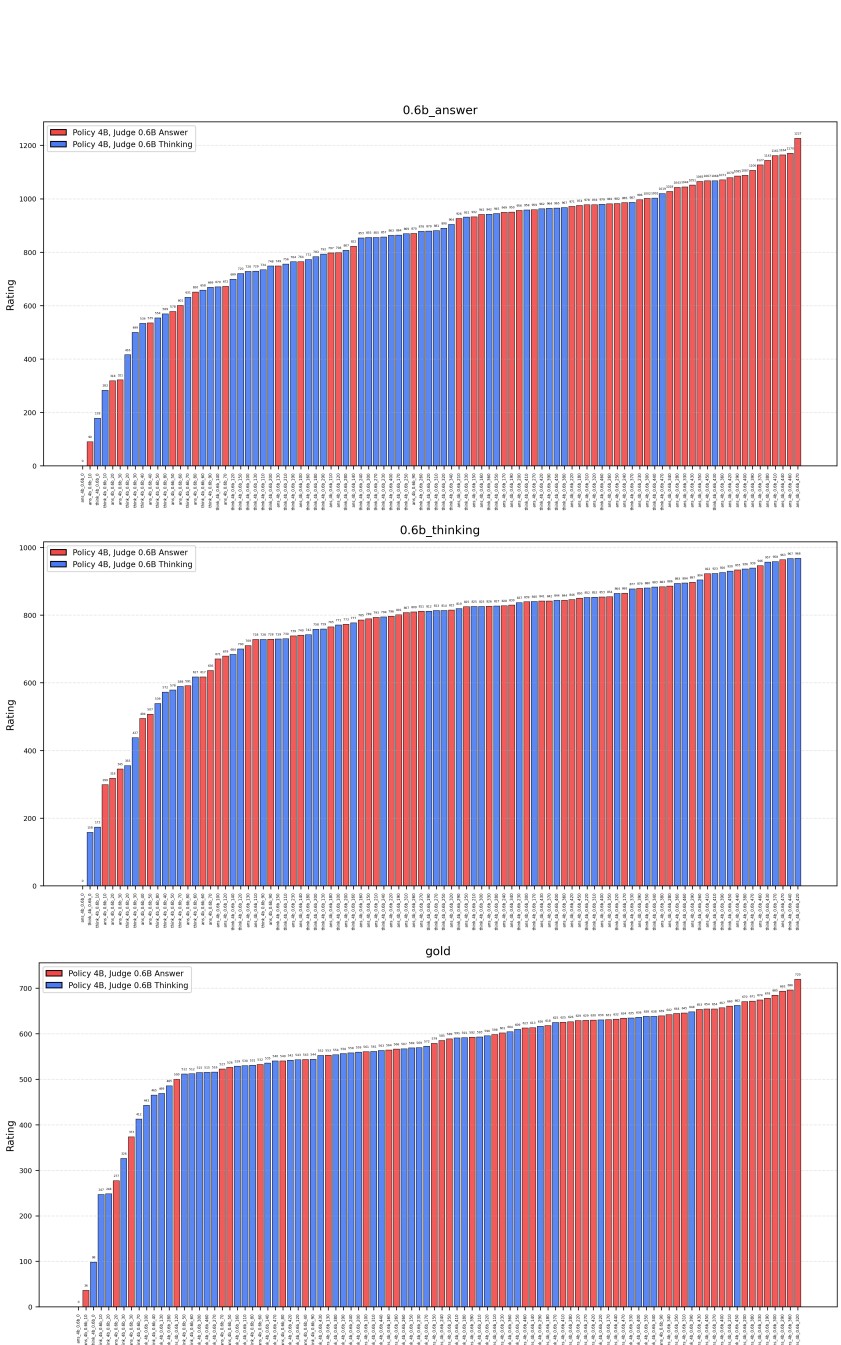

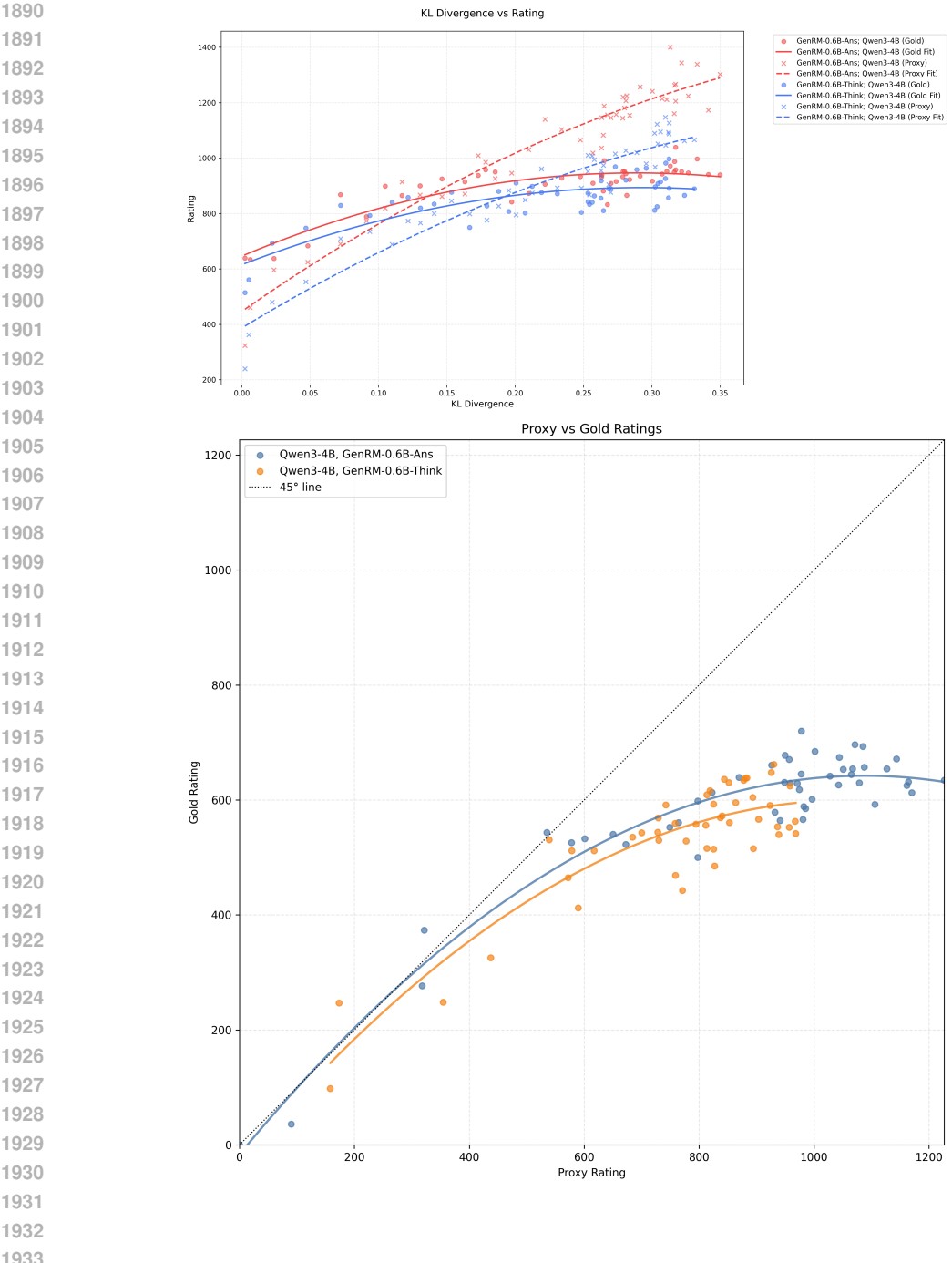

## C.12 THINKING VS ANSWER (4B VS 1.7B)

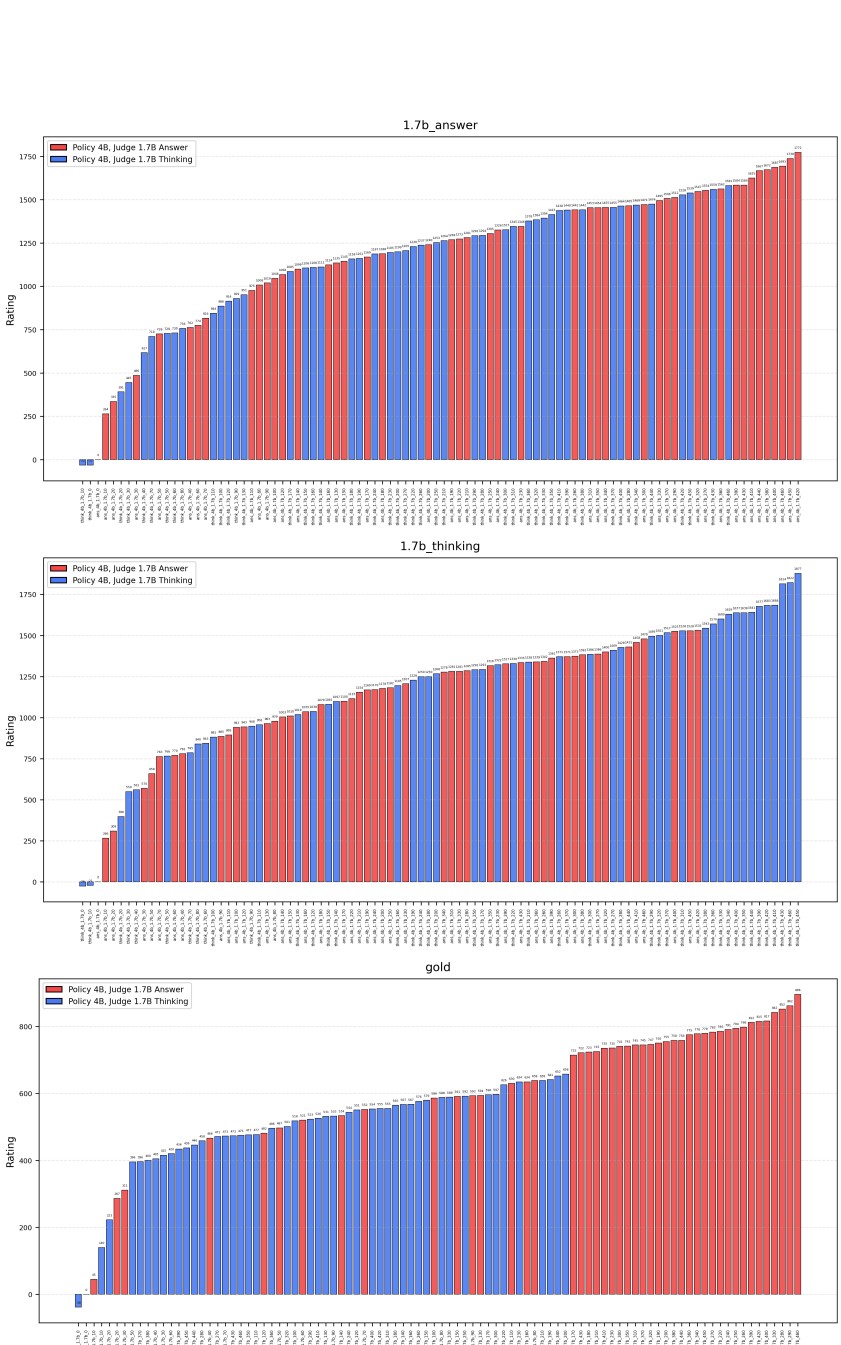

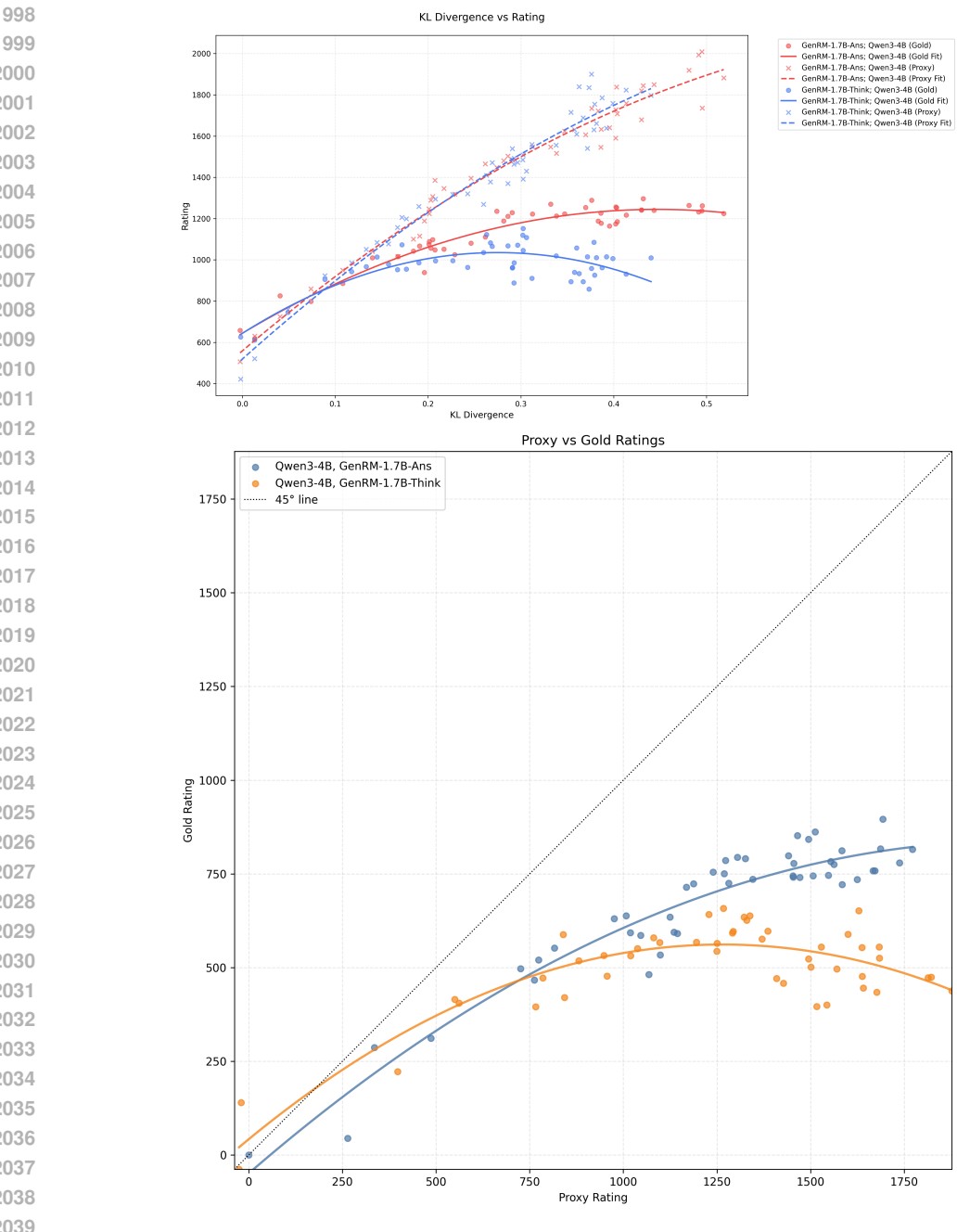

## C.13 THINKING VS ANSWER (4B VS 4B)

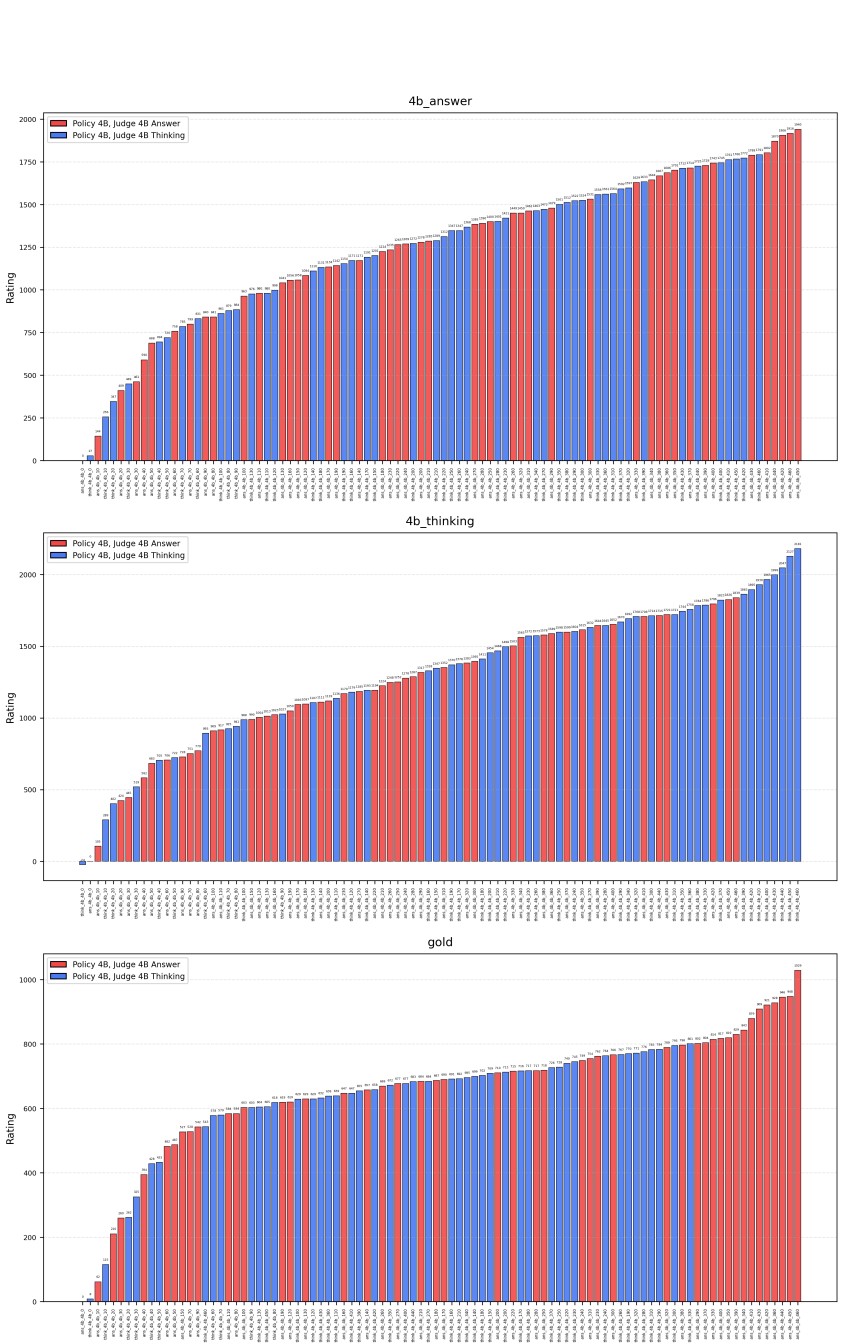

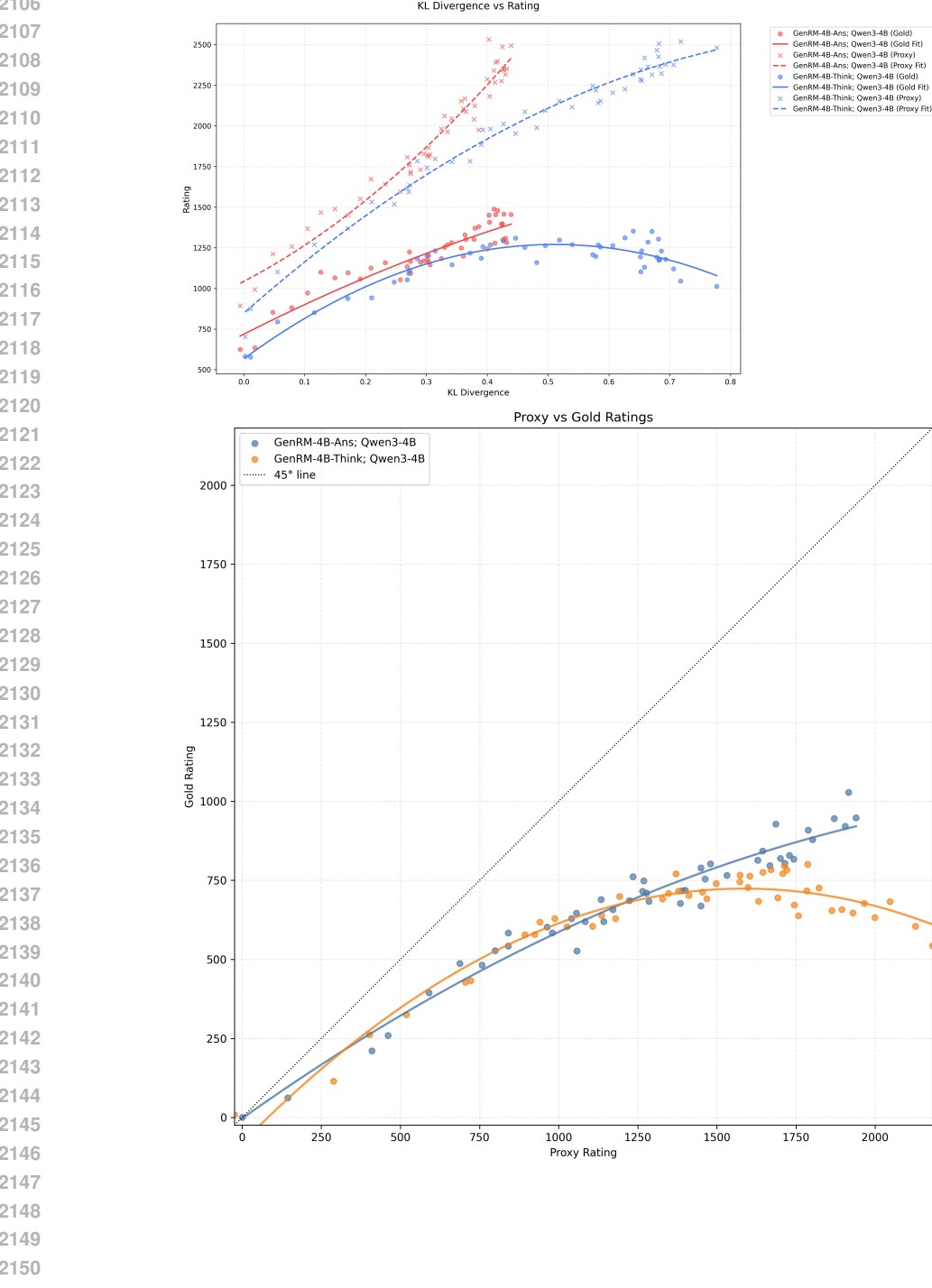

## C.14 THINKING VS ANSWER (4B VS 8B)

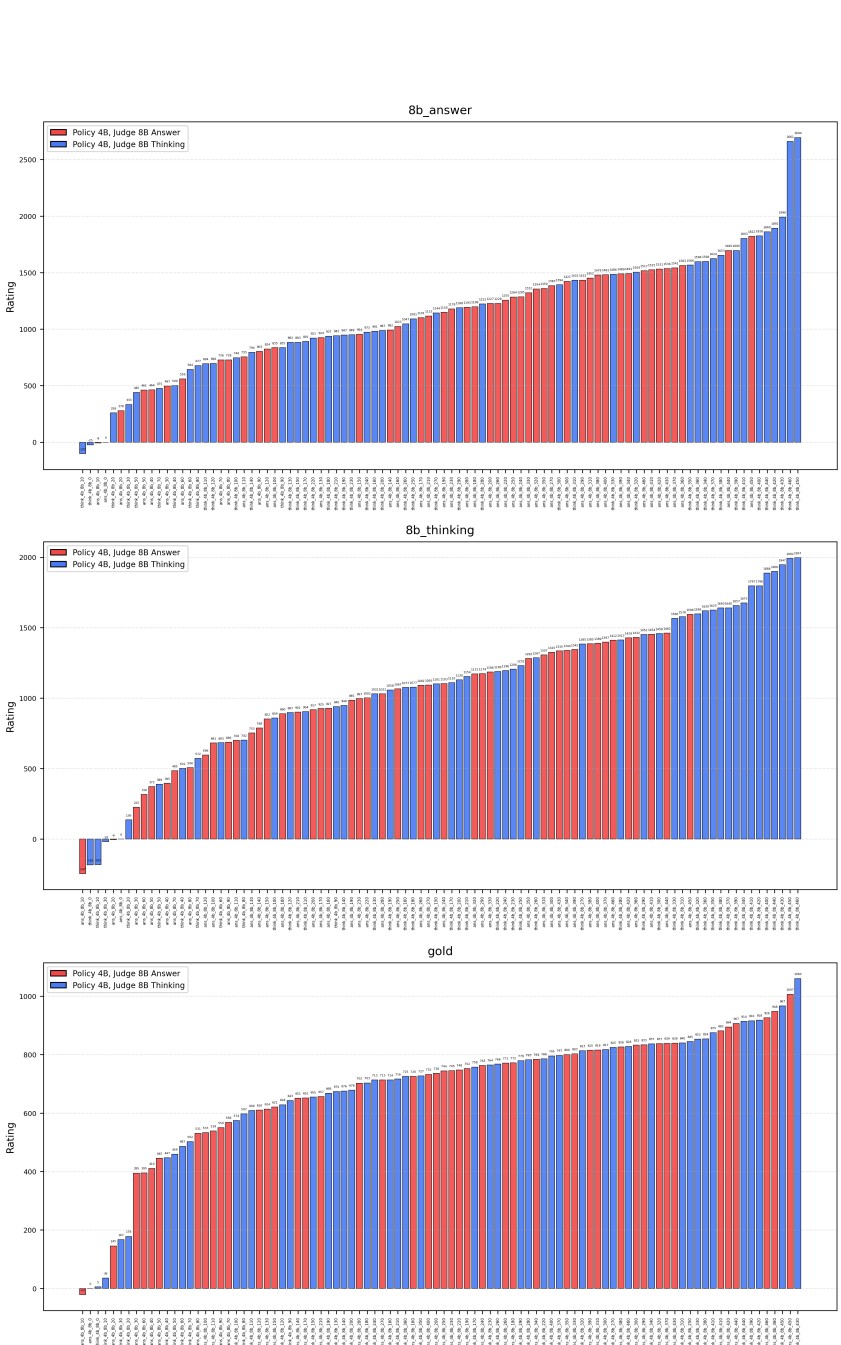

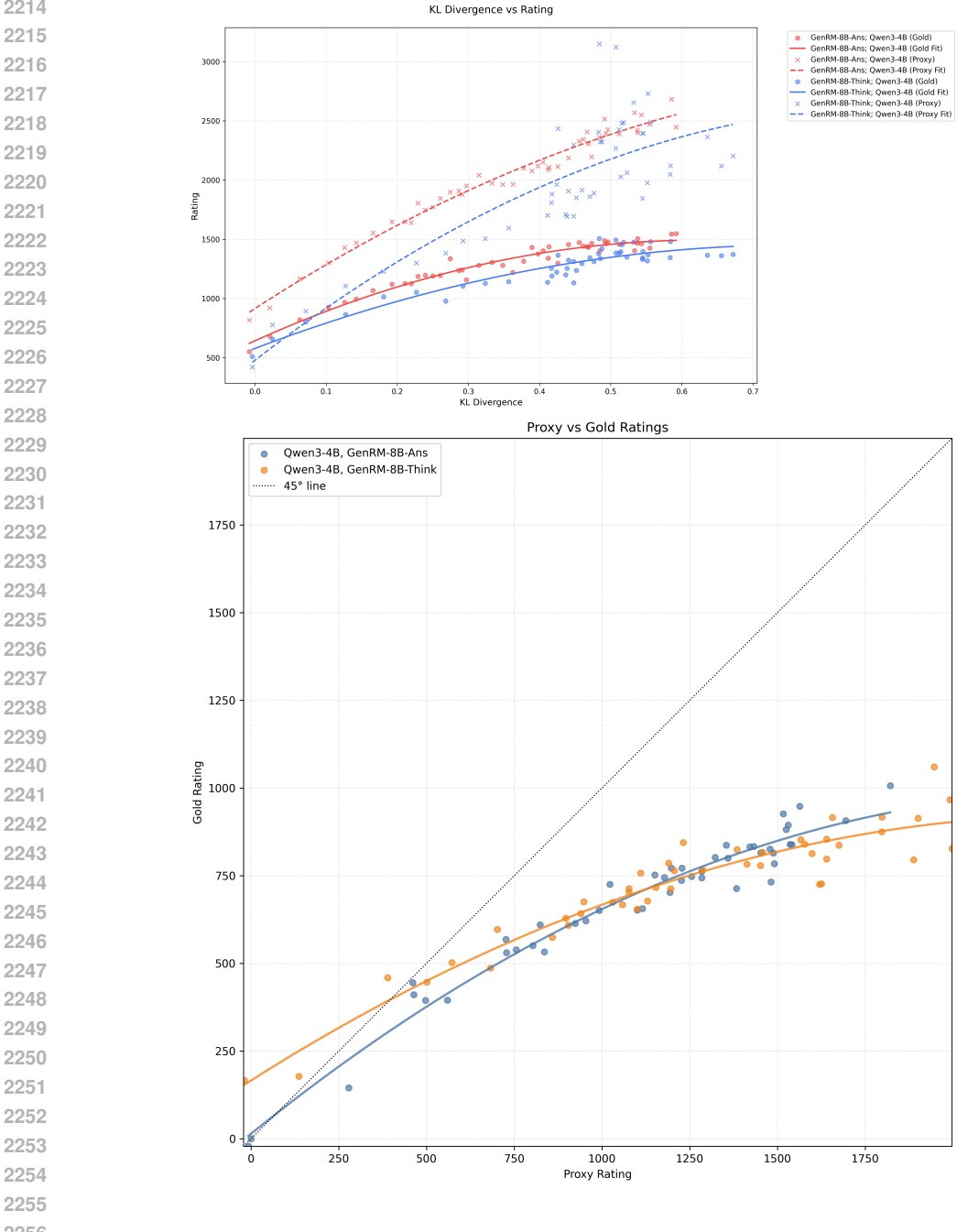

## C.15 THINKING VS ANSWER (4B VS 14B)

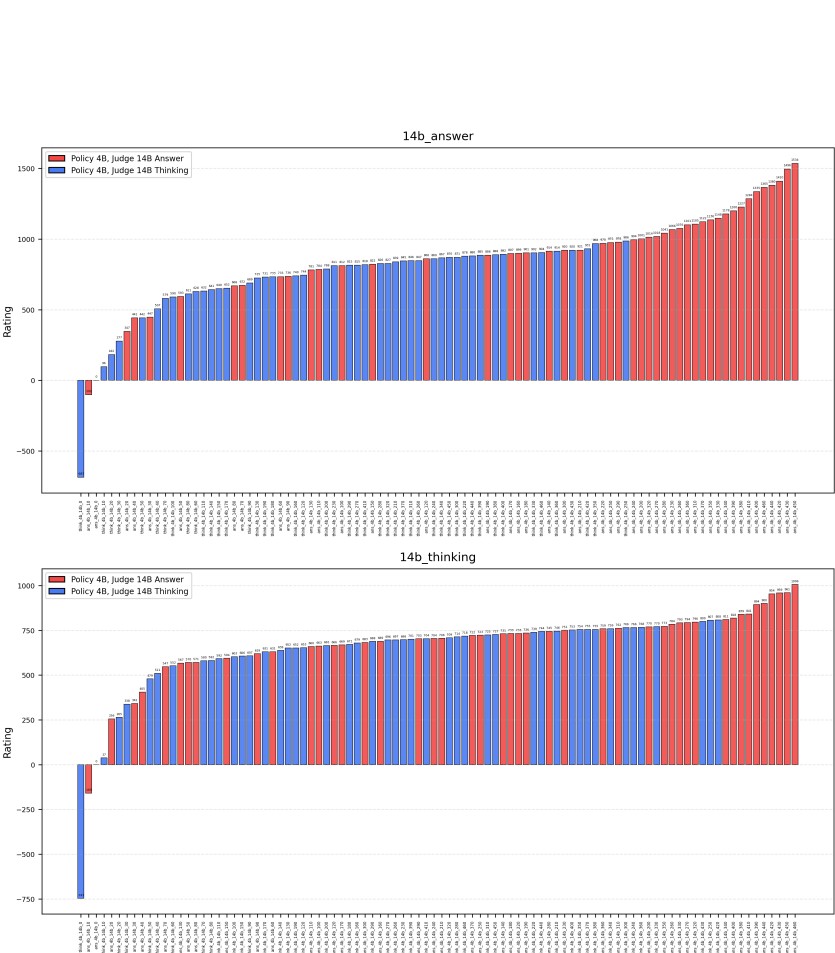

C.16   THINKING VS ANSWER (8B VS 4B)

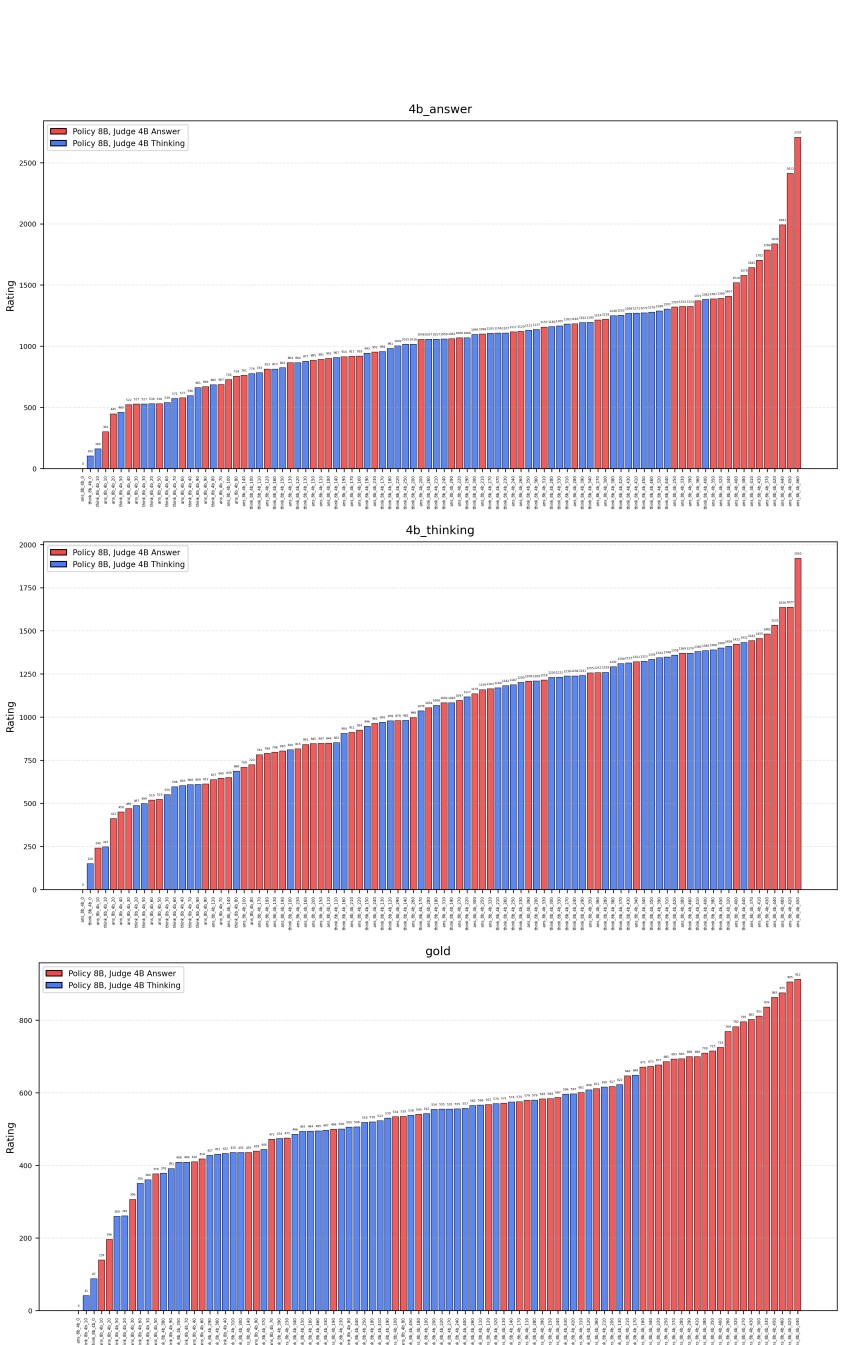

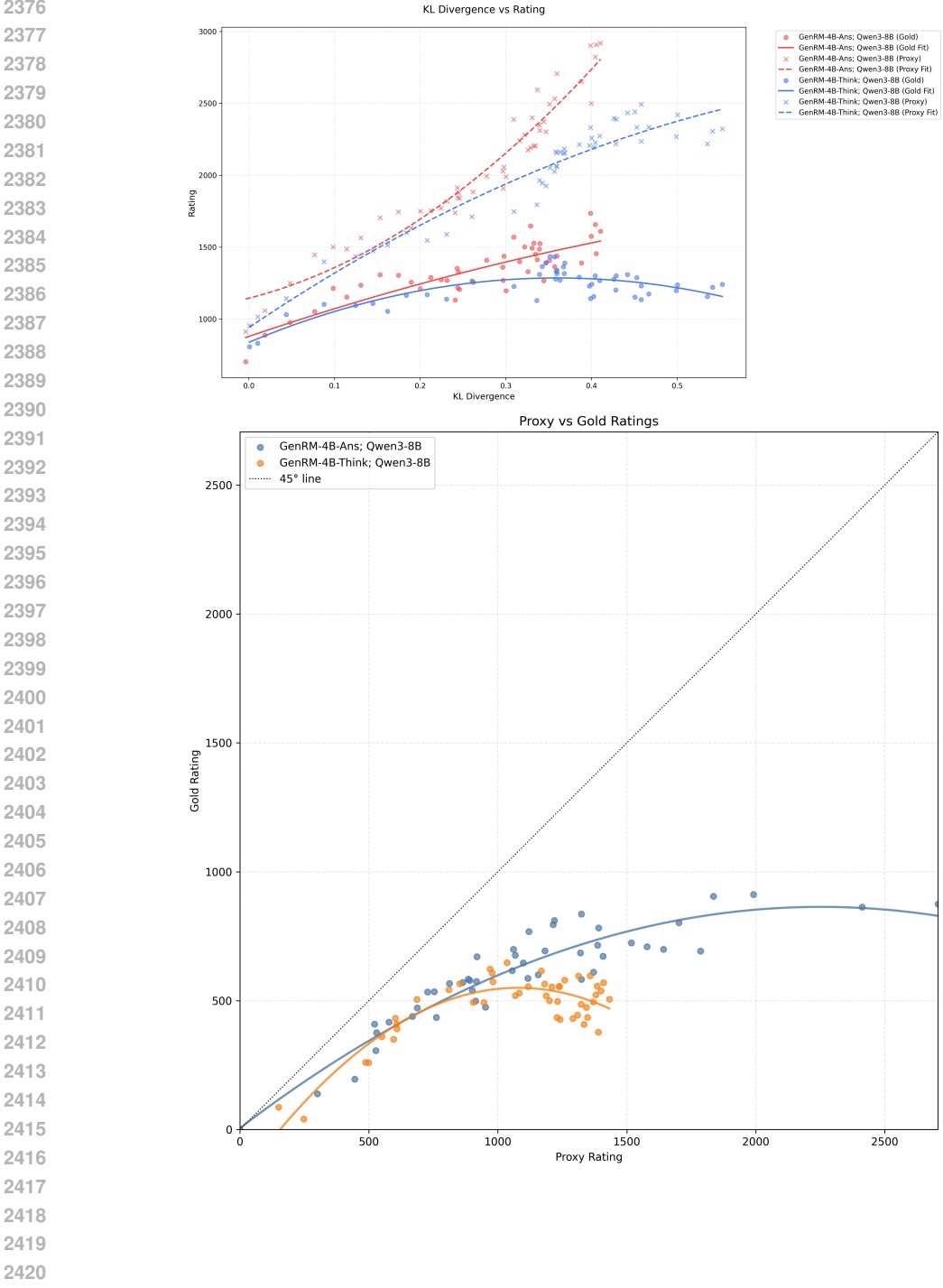

## C.17 TRAINED VS BASELINE

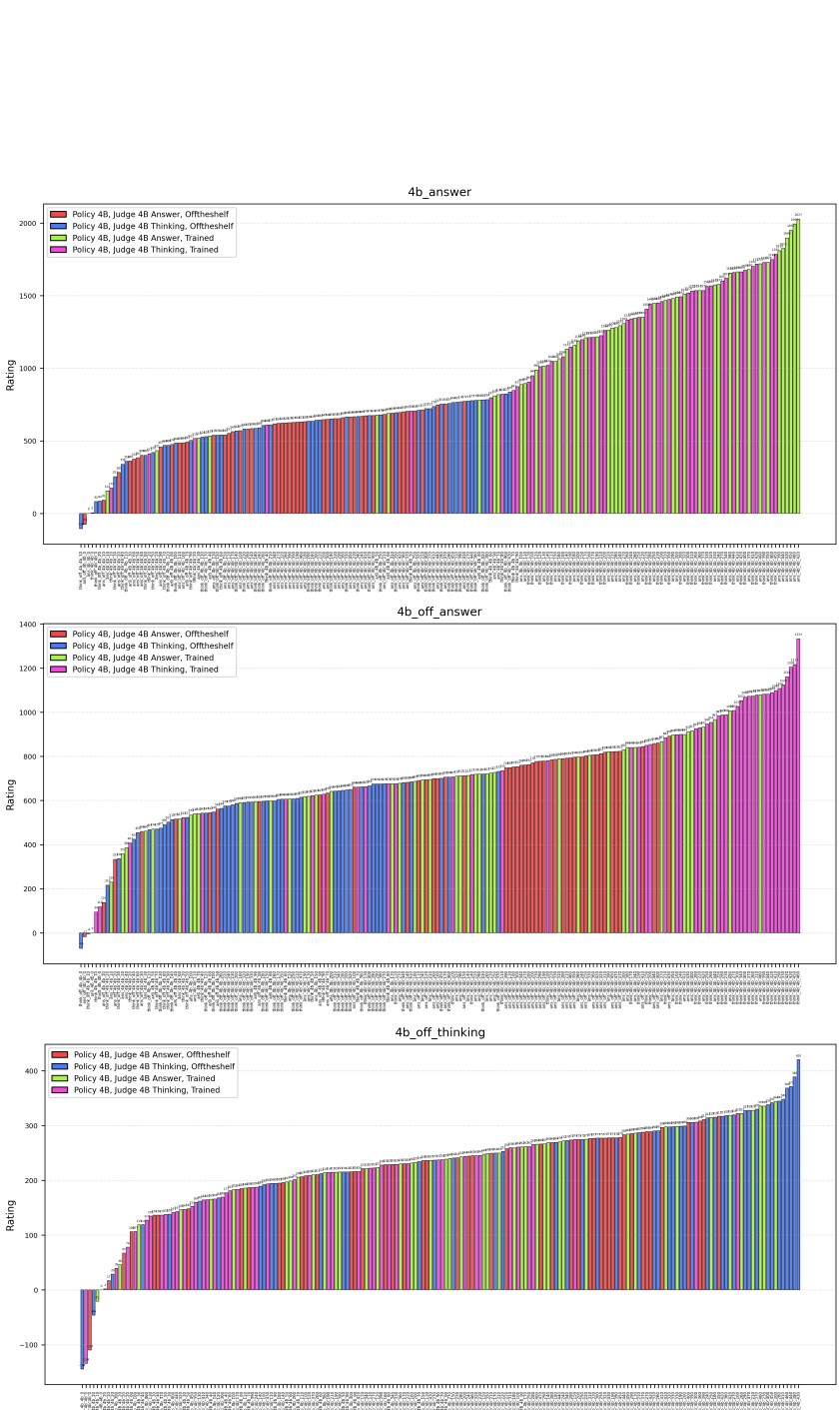

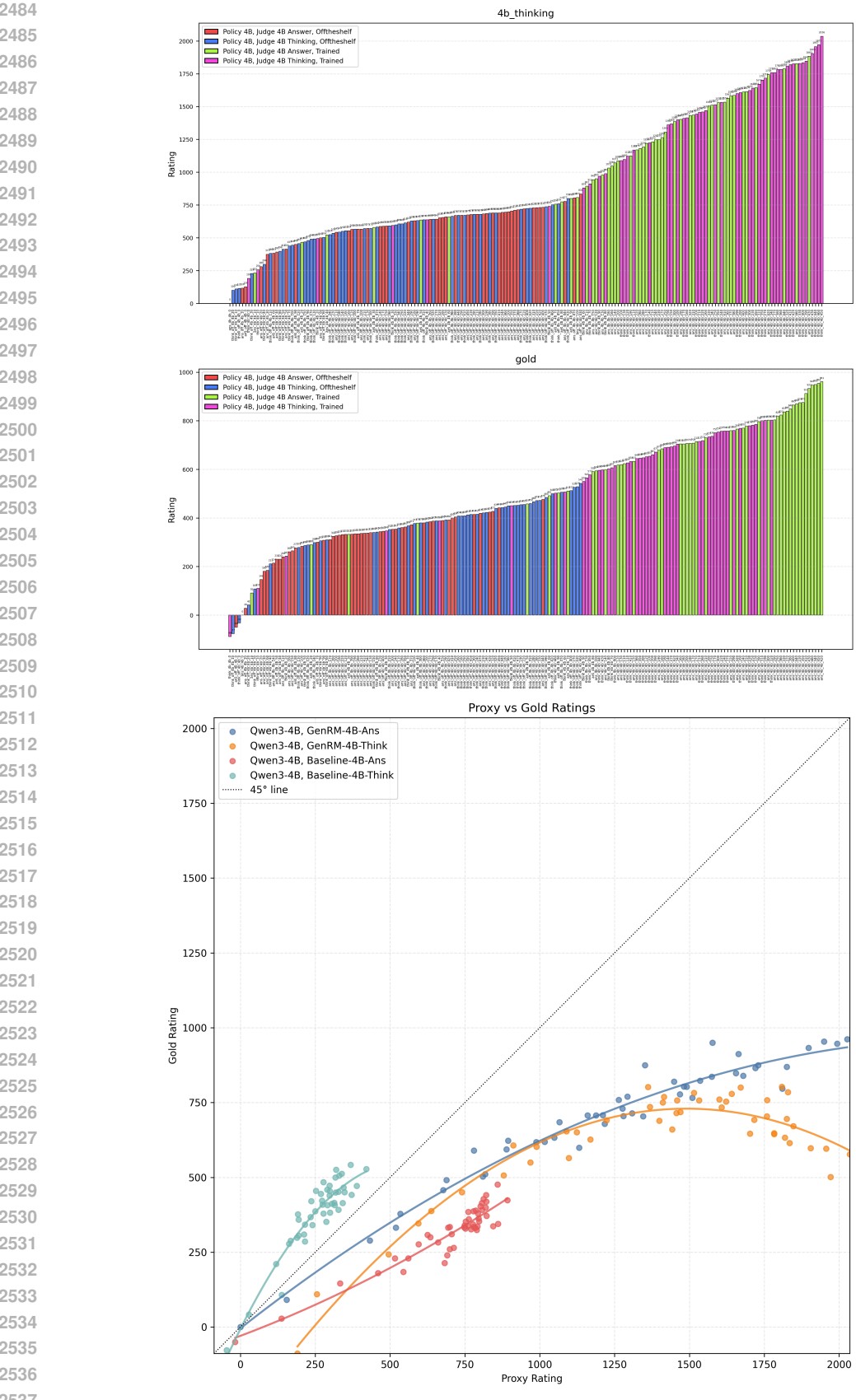

## C.18 ALL PARAMS

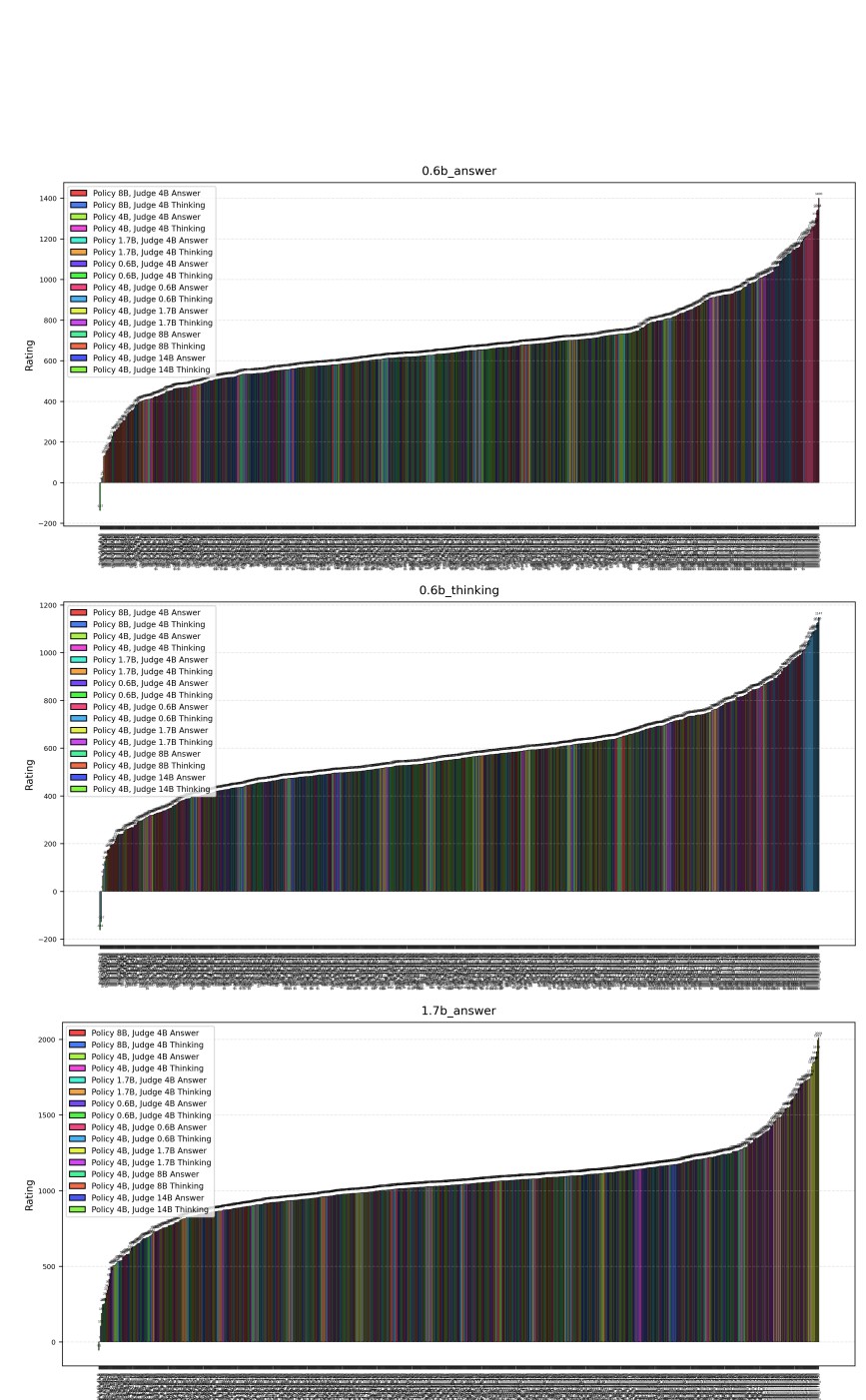

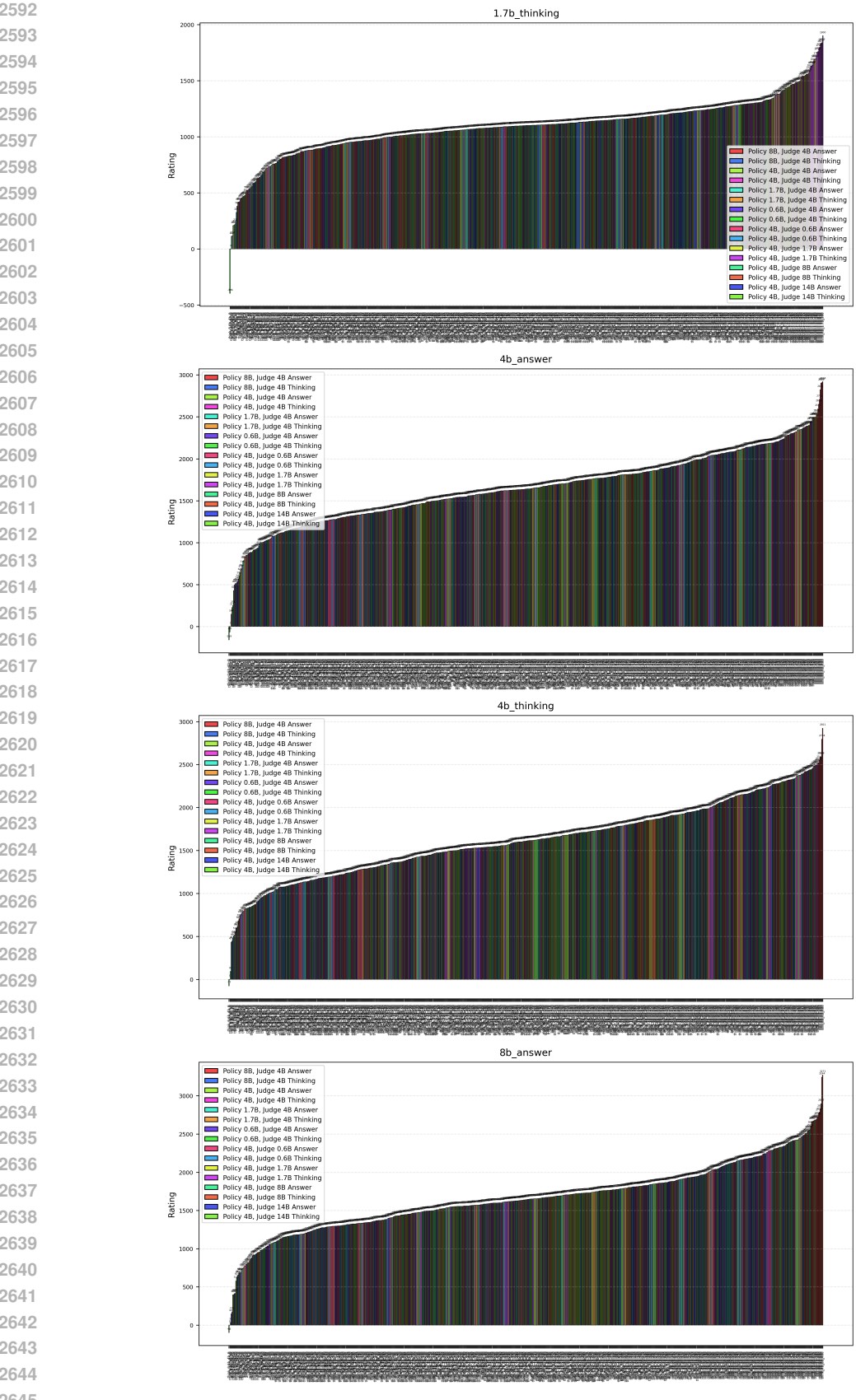

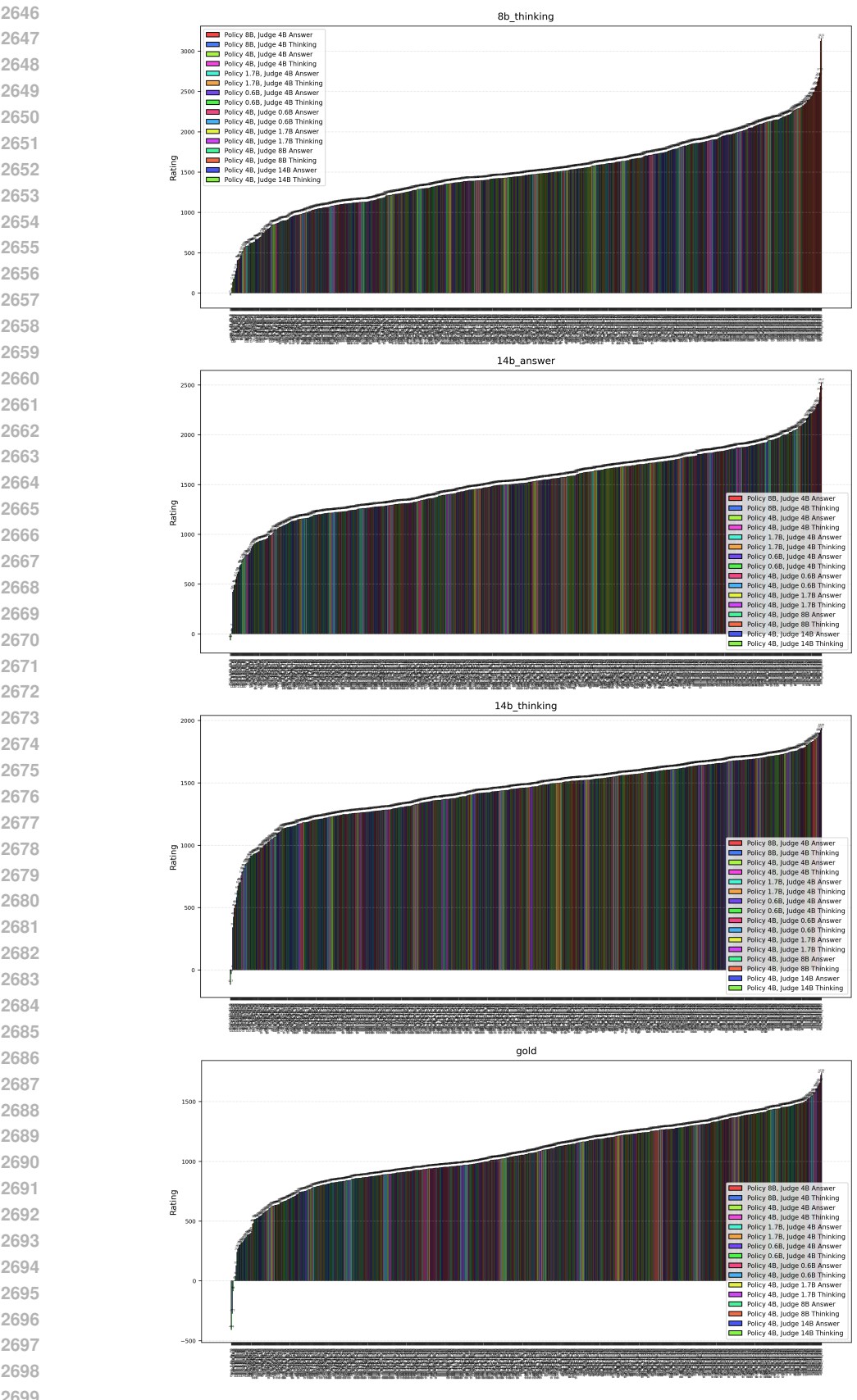

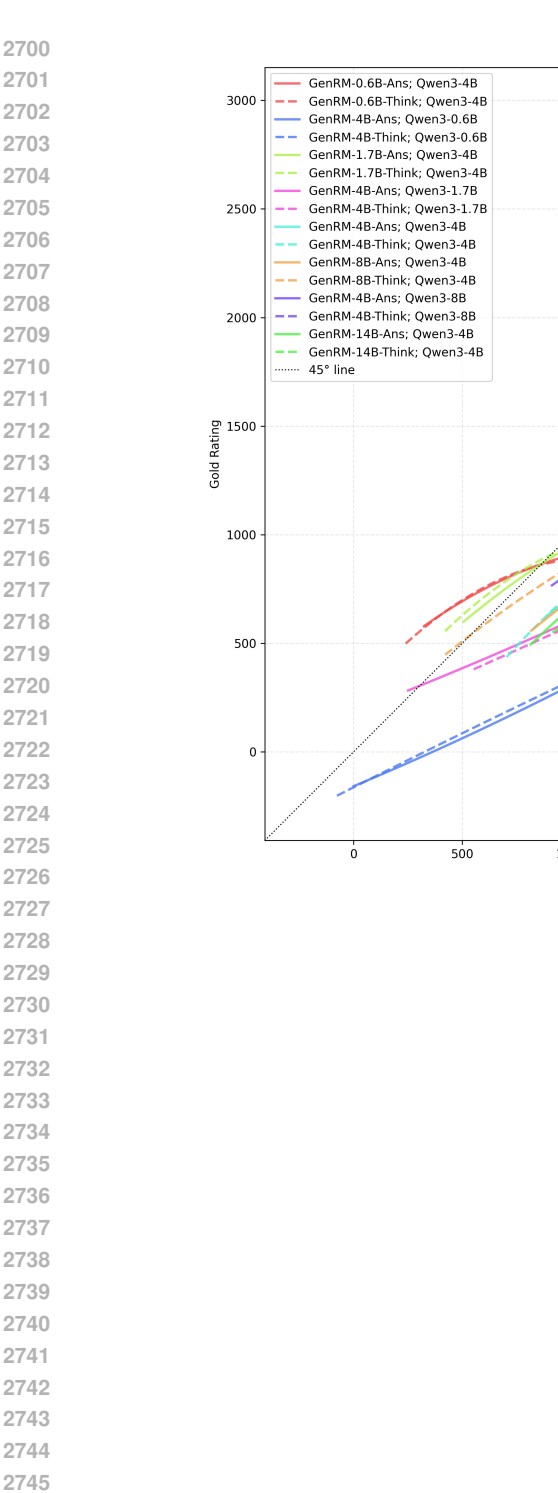

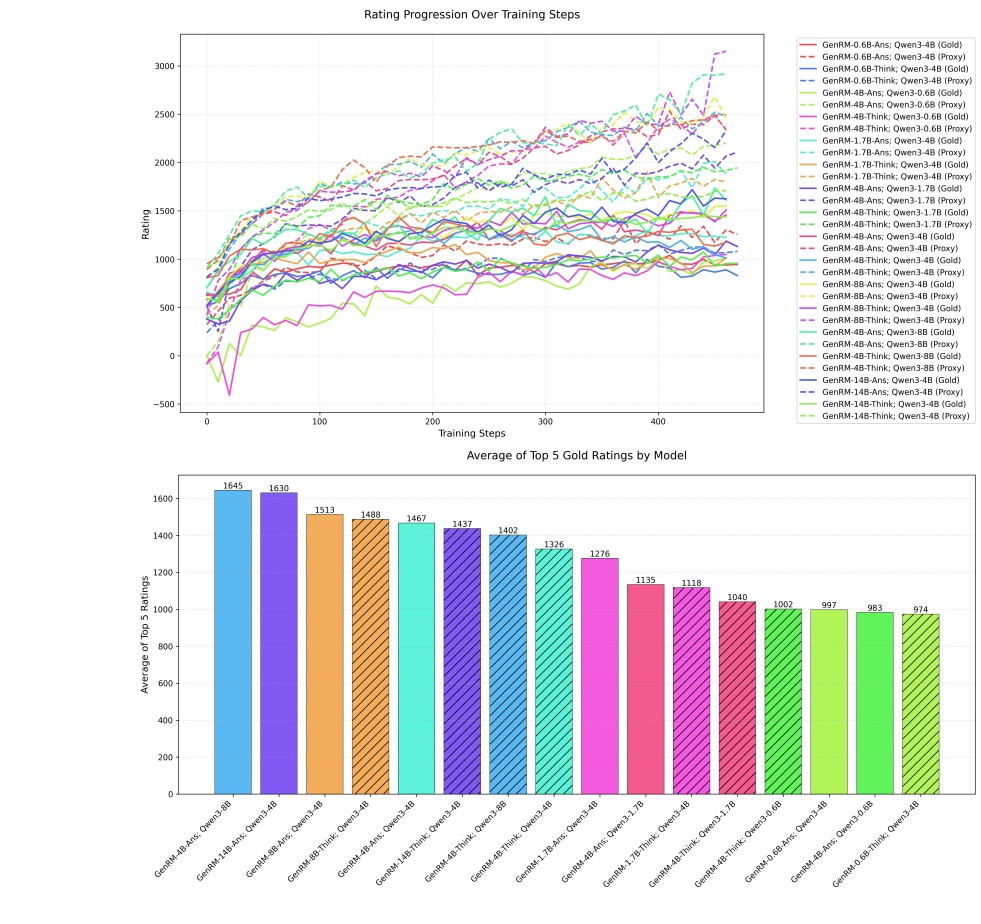

