# OpenReview forum: "Scaling Laws for Generative Reward Models"
_ICLR.cc/2026/Conference — Submitted to ICLR 2026_

### Official Review · Reviewer_WS1e · 2025-10-25

**Soundness:** 3
**Presentation:** 3
**Contribution:** 3
**Rating:** 8
**Confidence:** 4

**Summary:**

This paper studies the scaling laws of generative reward models, which outputs preference as tokens rather than scalar rewards. The authors studies various aspects, such as answer vs thinking, reward model size, policy model size, and the relationship between reward and policy model size. The main conclusion is that thinking GenRMs are better as verifiers, while answer GenRMs are better as rewards.

**Strengths:**

* The writing is pretty clear. The problem setup in section 2 is very easy to read, making it very clear even for reads not familiar with GenRM.
* The experiments are well conducted. The results are very clear and the authors summarized it very succinctly.

**Weaknesses:**

* A few things can be clarified which I list in the questions section.

**Questions:**

* Baselines refer to RMs that were not fine tuned, right? It's worth mentioning it in section 3 since the word "baseline" is almost never used in the paper until the results.
* What are the scatter points in the plots?
* How to actually read these plots? Typically in reward over optimization studies such as [1, 2], the x axis is KL, training steps, etc and the y axis is gold reward. However in your plots the x axis is proxy rating, which makes it a bit hard to interpret the plots.

[1] [Gao, L., Schulman, J., & Hilton, J. (2023, July). Scaling laws for reward model overoptimization. In International Conference on Machine Learning (pp. 10835-10866). PMLR.](https://arxiv.org/abs/2210.10760)

[2] [Rafailov, R., Chittepu, Y., Park, R., Sikchi, H. S., Hejna, J., Knox, B., ... & Niekum, S. (2024). Scaling laws for reward model overoptimization in direct alignment algorithms. Advances in Neural Information Processing Systems, 37, 126207-126242.](https://arxiv.org/abs/2406.02900)

---

> ### Author Response · Authors · 2025-12-03
>
> We thank the reviewer for the careful reading and for identifying concrete areas where our presentation could be clearer. Below we address each question and specify the revisions we will make.
>
> ---
>
> [Q1] Baselines
>
> Yes. By baselines we mean off-the-shelf Qwen3 models with no fine-tuning. In the revision, we will define “baseline” in §3 (Experimental Setup) and reiterate it in the first results subsection where the term appears, so readers do not need to infer it from figures.
>
> ---
>
> [Q2] Scatter plot points
>
> Thank you for pointing this out. We agree that the current description does not sufficiently explain what each point represents.
>
> Each dot is a single saved policy checkpoint from one online-DPO training run. Concretely, we train for ~460 steps and save a checkpoint every 10 steps, producing 47 checkpoints: (p_0, p_{10}, \ldots, p_{460}). Dots sharing a color correspond to the same (initial policy, proxy judge) configuration.
>
> To clarify how the axes are computed, consider Figure 2a as an example. We train the same 4B policy using four different proxy GenRMs (Answer/Thinking × off-the-shelf/trained), which results in (4 * 47 = 188) policy checkpoints.
>
> We then evaluate these checkpoints using a pairwise comparison procedure. We repeatedly sample two checkpoints and a random prompt from a held-out validation set of 1,000 prompts. Each policy produces one response, and the two responses are evaluated by five evaluators: the four proxy GenRMs and the Gold GenRM. This process is repeated for 20,000 comparisons.
>
> For each evaluator, we aggregate the comparison outcomes and compute an ELO score for all 188 checkpoints, yielding five separate ELO rankings. In the plots, the y-axis corresponds to the ELO score under the Gold GenRM, while the x-axis corresponds to the ELO score under the proxy GenRM that was used to train that policy. We will clarify this evaluation pipeline step by step in the revision.
>
> ---
>
> [Q3] How to read the plots
>
> The reviewer's intuition is correct that standard overoptimization plots (Gao et al., Rafailov et al.) show gold reward versus KL or training steps, which characterizes when collapse occurs.
>
> We intend these plots to visualize proxy–gold alignment during optimization. Training explicitly optimizes the proxy, so "good" training dynamics require that improvements in proxy-Elo also translate into improvements in gold-Elo. The dotted 45 line marks perfect alignment between proxy and gold rankings; deviations quantify misalignment/over-optimization directly.
>
> Our true objective is high performance under the Gold GenRM, which approximates human judgment. However, during training we do not have access to this gold evaluator. Training therefore relies on a proxy signal. The central question we study is how well optimization under the proxy aligns with true performance under the gold evaluator.
>
> Plotting proxy Elo (x) against gold Elo (y) directly visualizes whether gains under the training signal correspond to gains on the true objective. Deviations from the diagonal reveal where proxies mislead optimization.
>
> We agree that KL-based and step-based views provide complementary information. In the revision, we will add these plots to the appendix and explicitly justify our primary visualization choice in Section 4.
>
> ---
>
> Again, we appreciate the reviewer’s feedback and will revise the paper to improve clarity, terminology, and presentation so the evaluation and plots are straightforward for new readers.

---

### Official Review · Reviewer_c46P · 2025-10-31

**Soundness:** 2
**Presentation:** 1
**Contribution:** 1
**Rating:** 2
**Confidence:** 4

**Summary:**

This paper investigates how GenerativeRMs scale in reinforcement learning from AI feedback. In creative-writing preference tasks, ‘thinking’ GenerativeRMs trained with GRPO outperform ‘answer-only’ ones as evaluators, but during DPO training, ‘answer-only’ RMs train policies with higher Gold Elo than ‘thinking’ RMs across model sizes,.

**Strengths:**

**\[S1\] Relevance to Core Challenges in LLM Alignment**

The paper addresses a timely and important problem in LLM alignment, exploring how generative reward models scale and behave during policy optimization

**\[S2\] Comprehensive Empirical Examination.**

The work provides a broad empirical investigation across model scales and configurations, offering useful insights into the relationship between evaluator performance and policy optimization behavior.

**Weaknesses:**

**\[W1\] Misrepresentation of Background Knowledge**
The paper mischaracterizes the relationship between GenerativeRM and conventional reward models (scalar-headed RM). These approaches differ primarily in how they generate reward scores and chosen-rejected pairs, not in their fundamental learning processes. Both ultimately utilize the Bradley-Terry function as their optimization objective. Therefore, the authors' explanation of GenerativeRM in the introduction lacks technical accuracy and is not convincing.

**\[W2\] Insufficient Experimental Design**

While the authors aim to analyze the scaling laws of GenerativeRM, their experimental setup contains several significant limitations:

1. **Robustness of the gold evaluator**: As noted in their own limitations section, the gold evaluator is a fine-tuned Qwen3-32b model with only 79% accuracy. This creates a substantial risk that the generative RMs are being overoptimized to match this particular model rather than demonstrating true performance improvements.
2. **Lack of domain generalization**: The experiments are conducted to a single domain without OOD evaluation. This raises questions about whether the observed results would hold across different domains, limiting the paper's broader applicability.
3. **Limited optimization methods comparison**: The exclusive use of DPO represents a methodological weakness. Including alternative policy optimization functions such as PPO and BoN would establish greater fairness and strengthen the validity of their findings.

**\[W3\] Conceptual Issues with Comparative Analysis**
The comparison with answer-only GenerativeRM raises important conceptual questions. The core objective of GenerativeRM lies in its ability to use CoT reasoning / "thinking" capabilities to better evaluate responses. Training in an answer-only format effectively negates these advantages, making it conceptually indistinguishable from scalar-headed RM. A more meaningful analysis would include direct comparisons with conventional RM approaches to properly assess the claimed benefits of GenerativeRM.

**Questions:**

**\[Q1\]** DPO is fundamentally an offline methodology, so what is the justification for referring to it as "online DPO" in this work? This terminology appears inconsistent with the standard understanding of DPO.

**\[Q2\]** Line 152: The distinction between "human" and "gold" datasets remains ambiguous. The authors should provide precise definitions of these datasets, their composition, and the specific rationale for maintaining this distinction in the experimental protocol.

**\[Q3\]** Line 158-159: What’s the rationale behind re-annotating the LitBench using the gold model?

**\[Q4\]** Line 172-173: The response length appears quite short, which suggests responses might be getting cut off. Were there cases where responses were truncated prematurely? If so, this could significantly affect the evaluation quality, yet no analysis of this limitation was provided.

**\[Q5\]**  Which model was used as the policy in the experiments?

---

> ### Author Response · Authors · 2025-12-03
>
> We thank the reviewer for the careful reading and detailed feedback. These comments helped us identify several places where our presentation was unclear or imprecise. Below we clarify the intended technical setup and describe how we will revise the paper to improve accuracy and transparency.
>
> ---
>
> [W1] Misrepresentation of background knowledge
>
> Thank you for pointing this out. We agree that our introduction overstated the conceptual differences between Generative Reward Models (GenRMs) and conventional scalar-headed reward models.
>
> As you correctly note, these approaches do not differ in their fundamental optimization objective. Both ultimately rely on pairwise comparisons and a Bradley–Terry-style likelihood. The primary distinction lies in how preferences or reward scores are produced, not in the underlying learning principle.
> In particular, both approaches can:
>
> - Produce pairwise judgments (chosen vs. rejected). For scalar-headed RMs, this is typically done via a value head followed by a sigmoid; for GenRMs, this is often done by predicting an indicative verdict token.
>
> - Produce scalar rewards directly. Classical RMs do this via a scalar value head; GenRMs can do this indirectly through rubric-based aggregation, though this is less general and less intuitive for non-verifiable tasks.
>
> In the revision, we will correct the introduction to reflect this more accurately and clearly distinguish architectural/interface differences from optimization objectives.
>
> ---
>
> [W2.1] Robustness of the Gold evaluator
>
> Thank you for raising this concern. We believe there are two points that were not clearly communicated.
> First, our goal in introducing a Gold evaluator is not to perfectly replicate the human preference distribution, but to establish a stable, anchored preference distribution that can be queried consistently during policy training and evaluation. Human preferences serve as a natural anchor because they make the setup realistic, but the key requirement is consistency, not perfection. The central question we study is how well proxy GenRMs generalize when used to train policies relative to that anchor.
>
> Second, 79% agreement should not be interpreted as weak performance. On LitBench, prior work reports peak accuracies of roughly 78% even after extensive training across different architectures, model sizes, and training methods. Achieving 79% agreement therefore represents strong alignment with human preferences in this domain. For additional context, the OpenAI paper “Scaling Laws for Reward Model Overoptimization” reports Gold reward models with substantially lower agreement with human preferences (approximately 69.6%) that nonetheless serve as effective anchors for studying overoptimization dynamics. This supports the view that high (but imperfect) agreement is sufficient for a Gold evaluator to function as a meaningful and stable reference.
>
> In the revision, we will add this contextualization and explicitly clarify the role and expected performance level of the Gold evaluator so its use is not misinterpreted.
>
> ---
>
> [W2.2] Lack of domain generalization
>
> We agree that restricting experiments to creative writing limits generality, and we appreciate the reviewer calling this out. This was a known constraint during the study. Due to the computational cost of each training run, which ranges from 6 to 20 hours on eight H100 GPUs depending on configuration, we were unable to systematically repeat the experiments across additional domains.
>
> In the revision, we will explicitly emphasize this limitation and clarify that our conclusions are scoped to non-verifiable creative writing preferences, while outlining domain transfer as an important direction for future work.
>
> ---
>
> [W2.3] Limited comparison of optimization methods
>
> Regarding policy training, our choice of DPO is deliberate rather than incidental. GenRMs naturally produce pairwise preferences, not calibrated scalar rewards, and DPO allows us to use this signal directly without introducing ad-hoc scalarization.
>
> Regarding inference-time optimization, we agree that a Best-of-N comparison would be a useful addition. However, the primary gains in our setting come from policy training rather than inference-time scaling. Since our focus is on GenRMs as training signals, we did not include BoN comparisons in the main experiments. We will clarify this framing in the paper and list BoN comparisons explicitly as future work.
>
> In the revision, we will explicitly clarify the role of DPO as a preference-native policy optimization method,, and clearly state the scope of our evaluation focus. We will also note the exclusion of Best-of-N inference and list it explicitly as future work to avoid ambiguity for readers.

---

> ### Author Response · Authors · 2025-12-03
>
> [W3] Conceptual issues with comparing Answer-Only and Thinking GenRMs
>
> We agree that this distinction was not explained clearly enough in the paper, and we appreciate the opportunity to clarify it.
> Our study does not compare Generative Reward Models against classical scalar-headed reward models. Instead, we focus entirely on pairwise preference modeling and compare different ways of producing those preferences within a generative framework. In particular, we study how preferences over response pairs are computed, not whether to use pairwise preferences versus scalar rewards.
>
> Within this framework, an Answer-Only GenRM that emits a single verdict token (e.g., A or B) is architecturally and functionally equivalent to a scalar-headed reward model applied to a pair of responses. In both cases, the preference is derived from a single forward pass through the model, followed by a simple transformation of the final hidden state—either by reading logits for indicative tokens or by applying a linear layer with a sigmoid. Because these mechanisms are effectively equivalent, producing the preference as a token is simply a more natural choice in a language model, as it avoids modifying the architecture and keeps decision-making entirely in token space.
>
> Thinking GenRMs differ in a different dimension: they introduce additional inference-time computation by generating an explicit reasoning trace before emitting the final verdict. This corresponds to multiple serial inference steps rather than a single forward pass. The key question we investigate is therefore not architectural capability, but whether this additional reasoning structure and computation improves or degrades robustness when the model is used as a reward signal during online policy optimization, where response distributions evolve over time.
>
> In the revision, we will expand this explanation to clearly separate three concepts: (i) pairwise preferences versus scalar rewards, (ii) architectural equivalence between verdict tokens and scalar heads, and (iii) the role of inference-time computation and explicit reasoning. This should make the scope and intent of our comparison clear to readers.
>
> ---
>
> [Q1] Why “online DPO”?
>
> Thank you for noting the ambiguity. We agree this should have been defined more clearly.
>
> In offline DPO, response pairs are sampled once from a fixed reference policy and reused across many gradient updates, meaning the supervision distribution remains static throughout training. In contrast, in online DPO, response pairs are repeatedly resampled from the current policy after each update, so the supervision distribution evolves as the policy changes. This distinction is particularly important in our setting, as a central goal of the paper is to study off-distribution behavior as the policy drifts away from its initialization.
>
> While there are currently no theoretical guarantees showing that online DPO strictly dominates offline DPO, there is strong empirical evidence that online preference optimization can yield more stable and effective alignment dynamics when implemented carefully. In particular, the paper “Direct Language Model Alignment from Online AI Feedback” demonstrates clear empirical advantages of online preference optimization over offline variants (see Figure 3).
>
> In the revision, we will formally define online DPO, present the training objective explicitly in mathematical form, clearly distinguish it from classical (offline) DPO, and cite the relevant prior work so that the terminology and motivation are unambiguous to readers.
>
> ---
>
> [Q2] Human vs. Gold datasets
>
> Thank you for catching this ambiguity. The prompts and response pairs are drawn from the same LitBench distribution in both cases. The difference lies solely in the source of the preference labels. For the human dataset, labels come from LitBench annotations; for the Gold dataset, the same items are re-annotated by the trained Gold model.
>
> We will make this distinction explicit in the revised methodology section.
>
> ---
>
> [Q3] Why re-annotate LitBench with the Gold model?
>
> The reason is methodological clarity. If proxy GenRMs were trained directly on human preferences, we would lose the ability to independently measure where optimization breaks during policy training, since human preferences are not available on demand. By introducing a Gold model as a stable anchor, we can continuously evaluate policies and proxy GenRMs against a fixed preference distribution and clearly track over-optimization dynamics.
>
> We will restructure this explanation in the paper to make the motivation clearer.

---

> ### Author Response · Authors · 2025-12-03
>
> [Q4] Response truncation
>
> Response length is capped at 2048 tokens, which is large relative to typical Reddit stories. Truncation occurs in fewer than 2% of responses. When it does occur, truncated responses are almost always rejected in pairwise comparisons, so this behavior is not incentivized. We will add a short clarification noting truncation rates and their effect.
>
> ---
>
> [Q5] Policy models
>
> All policy models used in the experiments are from the Qwen3 family. We will state this explicitly for clarity.
>
> ---
>
> We sincerely thank the reviewer for the detailed and constructive feedback. These comments directly improve the clarity, correctness, and accessibility of the paper, and we will incorporate the above revisions to ensure future readers do not face similar confusion.

---

### Official Review · Reviewer_b7qy · 2025-10-31

**Soundness:** 3
**Presentation:** 2
**Contribution:** 3
**Rating:** 4
**Confidence:** 2

**Summary:**

The paper studies scaling laws for generative reward models (GenRMs) used as judges in preference-based alignment. Using Qwen3 models (0.6B–14B) and online DPO for policy training, the authors compare Answer-Only GenRMs (SFT, verdict token) to Thinking GenRMs (GRPO, rationale + verdict). On static, in-distribution evaluation, Thinking GenRMs modestly outperform Answer-Only (~1–2% accuracy). But during policy optimization, Answer-Only judges yield higher Gold Elo and appear more robust across sizes and $\beta$ settings; scaling the judge size consistently helps policies, often more than scaling the policy itself. The work introduces a unified Elo arena anchored to a Qwen3-32B “Gold” evaluator and reports rise-then-fall curves with $\sqrt(KL)$ consistent with over-optimization scaling laws.

**Strengths:**

- Clear evaluator vs. rewarder divergence: Thinking helps static judging but underperforms as a reward signal under matched budgets.
- Broad sweeps over judge/policy size and $\beta$; consistent PvG analysis and Elo reporting.
- Useful scaling-law framing (rise-then-fall vs $\sqrt(KL)$) and checkpoint analyses.

**Weaknesses:**

- Anchor bias / format asymmetry: The Gold evaluator is Answer-Only, which could favour Answer-Only proxies despite cross-judging. A thinking-style Gold or human adjudication would test robustness.
- Domain scope: Limited to creative writing; unclear if results extend to safety/helpfulness/dialogue or verifiable domains (math/code)
- Single backbone / algorithm: Only Qwen3 and online DPO are used; family/algorithm dependence remains open.
- Why thinking hurts remains mostly qualitative; more direct diagnostics would help.
- LLM usage clarity: It’s not fully clear whether any LLM assistance (e.g., for data filtering, rubric drafting, label consolidation, prompt generation) was used outside the main GenRM roles; if used, that should be documented for reproducibility.
- Several citations are missing in the related work part: (?). Please fix them.

**Questions:**

- Could you evaluate with a thinking-style Gold (or a human-anchored subset) to check if the Answer-Only advantage persists under a different anchor?
- How sensitive are results to inference-time compute for the thinking judge during training (e.g., k rationales, majority vote)? If you increase k during training under matched compute, does the gap close?
- Can you replicate a subset with a second family (e.g., Llama/Mixtral)?
- Please clarify any ancillary LLM usage (data prep, rubric creation, filtering, related work). If none, state explicitly; if yes, list models, prompts, and safeguards for leakage.

---

> ### Author Response · Authors · 2025-12-03
>
> We thank the reviewer for the detailed feedback and concrete suggestions. These observations highlight important design choices that deserve clearer justification. We address each concern below.
>
> ---
>
> [W1] Anchor bias / format asymmetry
>
> We investigated this concern empirically and found no evidence that an Answer-Only Gold evaluator biases results toward Answer-Only proxies. Two observations support this:
> 1) Static agreement does not favor answer-only at baseline. Off-the-shelf (untrained) answer-only and thinking judges show similar agreement patterns on the Gold-labeled set, with differences that are small relative to the trained-vs-baseline gap (Figure 1). At smaller sizes the Thinking models slightly outperform, at mid-size they are comparable, and at larger sizes Answer-Only holds a small edge. Overall, these differences are minor, suggesting no systematic advantage at the baseline level.
> 2) Pre-training alignment can favor thinking. In Figure 2a (baseline proxies), policies trained with thinking baseline judges trace a slope closer to the diagonal, indicating tighter proxy–gold agreement even though Gold is answer-only. In this setting, policies trained with Thinking proxies align more closely with the Gold Answer-Only evaluator than those trained with Answer-Only proxies. This suggests that prior to GenRM training, Thinking models are at least as compatible with the Gold evaluator as Answer-Only models.
>
> These observations suggest that the Answer-Only advantage we report in trained GenRMs reflects genuine optimization dynamics rather than evaluator format bias. We agree that a Thinking-style Gold evaluator would further strengthen robustness and will frame this explicitly as future work. For this reason, we explicitly list this as a limitation and we will add this analysis to Section 5 and strengthen the justification for our Gold evaluator design.
>
> ---
>
> [W2] Domain scope
>
> We agree that focusing on a single domain limits generality. We chose creative writing deliberately as an intentionally non-verifiable preference domain: correctness is stylistic and contextual rather than programmatically checkable, so static evaluator accuracy can saturate while reward hacking and proxy–gold divergence remain salient. This setting directly stress-tests the paper’s central question: whether stronger static evaluators necessarily make better in-loop reward signals under distribution shift induced by policy optimization.
>
> We were also acutely aware of the domain-scope limitation throughout the study. Replicating our full experimental grid in additional domains would scale cost roughly linearly: each (judge, policy) training configuration requires an online DPO run plus Elo evaluation and costs ~6–20 hours on 8×H100 (not including judge training and cross-evaluation). Given the number of ablations we run (policy size × judge size × judge mode × training-budget/β/checkpoint sweeps), a full multi-domain replication would require multiple additional thousands of H100-hours.
>
> To address this within constraints, we ran preliminary cross-domain checks on OpenOrca using off-the-shelf Qwen3 policies/judges/Gold evaluator. These preliminary results support the same qualitative pre-training trend observed in the main domain (thinking judges start stronger as static evaluators), and we will include them in the appendix clearly labeled as preliminary. In the revision, we will also (i) explicitly justify the domain choice earlier in the paper and (ii) foreground domain generalization as an open question.

---

> ### Author Response · Authors · 2025-12-03
>
> [W3] Single backbone and optimization algorithm
>
> We agree that restricting experiments to a single model family and optimization method limits generality; we discuss this dependence in §5 and will surface the key rationale earlier (in §3) so readers do not have to infer these design constraints from the discussion.
>
> Qwen3 is currently the only open-weight family satisfying both requirements for our study: (1) native support for hybrid thinking and non-thinking modes on the same checkpoint via template flags, and (2) a broad parameter ladder of six dense sizes spanning 0.6B to 32B. Establishing scaling laws requires systematic variation across model scales under controlled conditions, and the Qwen3 family lets us vary judge mode and scale while holding backbone architecture and instruction-tuning distribution fixed. The closest open-weight alternatives typically satisfy only one of these requirements, either supporting dual-mode behavior at a single parameter size or releasing multiple sizes without a supported mode toggle. Using such alternatives would require custom training to induce comparable modes across scales, introducing confounds.
>
> For the Gold evaluator, our priority was ensuring a meaningful preference gap between Gold and untrained proxy GenRMs at initialization. Fine-tuning Qwen3-32B on human preferences achieved this: Figure 1 shows a ~20% accuracy gap between trained and off-the-shelf proxies when evaluated against Gold labels, confirming substantial room for optimization. A cross-family Gold evaluator could further reduce shared inductive bias but is orthogonal to our central contribution and represents valuable future work.
>
> For policy models, restricting to a single backbone allows us to cleanly isolate the interaction between policy size and GenRM size, which is the focus of our scaling analysis.
>
> Regarding optimization, we study GenRMs which produce pairwise preferences as their native output, a structure that naturally lends itself to direct alignment methods consuming preference pairs. Online DPO, where preferences are collected on-policy during training, consistently outperforms offline variants when implemented carefully (Guo et al., 2024, Figure 3). We will make this connection between GenRM output format and algorithm choice explicit in §2 and cite the relevant prior work.
>
> ---
>
> [W4] Why thinking hurts
>
> We agree the current discussion is qualitative and will restructure it to separate evidence from hypotheses.
>
> Empirical finding:
>
> - Higher in-distribution judge accuracy does not reliably imply better in-loop training. Figure 2b already shows a key symptom: as the Thinking judge continues to improve on validation accuracy, policy Gold-Elo peaks at an intermediate judge checkpoint and then declines, consistent with increasing exploitability/overfitting under distribution shift.
>
> Mechanistic hypotheses:
>
> - One key observation is that higher in-distribution GenRM accuracy does not necessarily translate into better policy training. Since Online DPO changes the response distribution during training, GenRM robustness across the full domain is more important than performance on a fixed validation set.
>
> - Figure 2b provides evidence for this. We trained policies using multiple checkpoints of a Thinking GenRM during GRPO training. While GenRM accuracy steadily improves on the in-distribution set, policy performance peaks at an intermediate checkpoint and then degrades. This suggests overfitting to the evaluation distribution, hurting generalization during policy optimization.
>
> - Another possibility is that Thinking GenRMs are less robust to distribution shifts. GRPO optimizes final preferences rather than reasoning correctness, allowing flawed reasoning paths that still reach correct labels. These weaknesses may be tolerated in-distribution but exploited out-of-distribution. Since Thinking models split computation between latent activations and explicit language traces, this may introduce additional failure modes compared to Answer-Only models, which perform computation entirely in the latent space.
>
> In the revision, we will make these hypotheses explicit and add the most direct diagnostics we can support from existing logs (e.g., positional inconsistency rates), so the explanation does not rely on qualitative intuition alone.
>
> ---
>
> [W5] LLM usage clarity
>
> Thank you for flagging this. Beyond the GenRM and policy models described in the method, we did not use LLMs for dataset filtering, rubric creation, label consolidation, or prompt generation. We will add an explicit reproducibility statement and include all policy/judge prompts in the appendix.
>
> ---
>
> [W6] Missing citations
>
> We will correct all missing references in the revision. Thank you for catching this.

---

> ### Author Response · Authors · 2025-12-03
>
> [Q1] Thinking-style Gold or human-anchored evaluation
>
> As discussed in W1, the current evidence suggests that if any bias exists, it favors Thinking proxies rather than Answer-Only ones. We agree that repeating the experiments with a Thinking Gold evaluator would further strengthen the results. However, doing so would require retraining proxy GenRMs and policies, which is not feasible within our current computational budget. We will frame this explicitly as future work.
>
> ---
>
> [Q2] Sensitivity to inference-time compute for thinking judges
>
> We did not run targeted experiments varying inference-time compute during policy training. However, Figure 1 shows that even large majority votes provide only marginal gains in in-distribution accuracy. This motivated our decision not to pursue these experiments further. We agree that out-of-distribution effects could differ and consider this an interesting direction for future work, which we will note explicitly.
>
> ---
>
> [Q3] Replication with another family
>
> As noted in W3, no other public open-weight model family currently supports both thinking and non-thinking modes across a range of sizes. This constrained our design choices, and we will clarify this limitation more explicitly.
>
> ---
>
> [Q4] Ancillary LLM usage
>
> All LLM usage has already been specified in the paper. As mentioned above, we will include policy and GenRM prompts in the appendix to make this fully explicit.
>
> ---
>
> We thank the reviewer again for the careful and constructive feedback, which will directly improve clarity, transparency, and reproducibility in the revised paper.

---

### Official Review · Reviewer_UekM · 2025-10-31

**Soundness:** 1
**Presentation:** 2
**Contribution:** 2
**Rating:** 4
**Confidence:** 4

**Summary:**

This paper investigates how Generative Reward Models (GenRMs) behave when used for reinforcement learning from AI feedback (RLAIF), particularly as replacements for traditional Bradley–Terry scalar reward heads. The authors study GenRMs trained to “think” (i.e., generate rationales before verdicts) using GRPO versus simpler answer-only versions trained with supervised fine-tuning (SFT), across varying scales of policy and reward models (0.6B–14B parameters) in the Qwen3 family. Results indicate that answer-only models can be more effective as evaluators.

**Strengths:**

Authors are tackling a relevant and interesting topic, and the experiment suite is abundant in ablations. It is particularly encouraging to see work on generating reward models on hard to verify environments and while I have concerns with the current state of the work I want to encourage the authors to keep pressing on this.

**Weaknesses:**

My main concern is, while the work is promising, I don't see that the experiments support the main conclusions that authors draw. Particularly:

* Prior to training, thinking models perform better than the answer-only models. Then authors perform two distinct training procedures for answer only models (SFT) and thinking ones (GRPO), an  then they test them with in-distribution data. Thus, this is not an apples-to apples comparison.  To get the conclusions valid authors should have tested both methods with the same training technique (or show evidence that each kind of model works better with a different one) and then test them with a new data distribution. Note that for the baseline models this was an out-of-distribution set. I think there is worth on the results that authors currently have, but the claims don't align.

* it was very contradictory where authors say they want to analyse the presumption that larger models reward better but the whole experiment is based on having a bigger model as gold evaluator

* It was not very clear what the ablations where trying to accomplish sometimes, would have been beneficial.

* Some broken references at the related lit section

**Questions:**

* What was the test-train split done on the original dataset?

---

> ### Author Response · Authors · 2025-12-03
>
> [W1] Different training procedures for answer-only and thinking models
>
> Thank you for raising this concern. We understand why our current phrasing may suggest that an “apples-to-apples” architectural ablation – training both answer-only and thinking models under identical algorithms – would be necessary. Our comparison, however, is not an architectural study but a comparison of the best-supported pipelines that practitioners actually use today.
>
> 1. Why the comparison remains valid even with different training techniques.
>
> For each model format, we apply the post-training method that empirically performs best:
> For answer-only GenRMs, SFT is the most appropriate and highest-performing method. In a binary A/B task, the supervision signal is unambiguous, and SFT directly optimizes the correct target.
>
> For thinking GenRMs, the supervision problem is fundamentally different: a chain-of-thought has no single correct trajectory. Current practice, supported by recent work, overwhelmingly relies on RL-based GRPO-like objectives.
>
> To avoid simply assuming this, we ran an explicit ablation on our 4B thinking judge, all evaluated on a held-out split of the Gold-labeled dataset:
> - Off-the-shelf model: 60.0%
> - Distillation from Qwen3-32B thinking traces (SFT-style): 64.7%
> - STaR: 60.7%
> - GRPO with accuracy + positional consistency rewards: 83.2%
>
> Only GRPO yields a large improvement; the SFT-style objectives give modest or no gains. This is why, in the main experiments, we treat “SFT answer-only vs. GRPO thinking” as the best-supported pipelines for each judge type, not arbitrary asymmetry.
>
> Under this framing, the comparison is apples-to-apples at the pipeline level: we compare the best-supported answer-only pipeline against the best-supported thinking pipeline, trained on the same data and evaluated under the same conditions.
>
> 2. Why a symmetric SFT/GRPO study across both model types is not meaningful
>
> Applying both SFT and GRPO to both architectures would not result in a cleaner comparison; instead, it would introduce ill-posed or unnatural training setups.
>
> SFT on thinking models is under-defined. It requires committing to a single “correct” chain-of-thought, which introduces additional arbitrary design choices and weakens comparability rather than improving it.
>
> Conversely, GRPO on answer-only models is not meaningful. Without intermediate trajectories to optimize, GRPO collapses to essentially the same signal as SFT, but with added variance and additional hyperparameters.
>
> Training both architectures under the same algorithm therefore does not produce a fairer comparison; it forces each into regimes they are not designed for. Answer-only and thinking models are fundamentally different objects, and the most faithful comparison is between each model trained using the method it performs best under.
>
> 3. What our study actually isolates.
>
> Both model types are trained on the same dataset and evaluated on the same held-out split. Before training, thinking models outperform answer-only models; after training, they retain a slight static advantage as evaluators. The key result is that, despite this, answer-only judges yield higher Gold Elo and smaller proxy–Gold gaps when used as online reward models during policy optimization.
>
> We will revise the paper to clarify this framing and avoid the impression that our goal was a purely architectural ablation.

---

> ### Author Response · Authors · 2025-12-03
>
> [W2] Apparent contradiction regarding model scale and the Gold evaluator
>
> Thank you for pointing this out. We see how the current phrasing could suggest a contradiction, and we will clarify the intent more explicitly.
>
> In principle, the ideal reference distribution for evaluating reward models would come from human preferences. However, evaluating every policy checkpoint with human annotators is not feasible in an online RL setting. For this reason, we adopt a common and practical approach: using a fixed “Gold” model as a surrogate for human judgment, following the methodology introduced in “Scaling Laws for Reward Model Overoptimization” (OpenAI). Their work likewise uses a larger Gold model (6B) to anchor the behavior of significantly smaller models (3M–3B).
>
> In our setup, the Gold evaluator serves only as a fixed reference distribution for Elo comparisons. It is never trained further, optimized against, or included in any of the scaling analyses. Our scaling results concern only proxy GenRMs and policies (0.6B–14B, answer-only vs. thinking), all evaluated relative to the same constant anchor. We do not compare the Gold model itself to smaller models; rather, we compare smaller models to one another under a consistent reference signal.
>
> We chose a larger model as Gold because it is more likely to approximate human preferences than a smaller one; but even if the approximation were imperfect, it would not affect the validity of the comparisons as long as the anchor is held constant.
> In the revision, we will edit the introduction and methods sections to make this distinction explicit and remove any impression of internal inconsistency.
>
> ---
>
> [W3] Purpose of ablations
>
> Thank you for flagging this. We will revise the experiments section and figure captions to make each ablation’s purpose explicit. Concretely:
> - For each main figure (GenRM size sweep, policy size sweep, β sweep, amount of GenRM training), we will add a one-sentence “takeaway” in the caption (e.g., “Fig. 4: increasing GenRM size improves Gold Elo and delays over-optimization, especially for answer-only judges”).
> - In the text, we will add a short paragraph before each subsection explaining which question the ablation is answering (e.g., “Does scaling the judge help more than scaling the policy?”).
>
> These edits are purely expository and do not change any experiments.
>
> ---
>
> [W4] Broken references
>
> We will fix all “(?)” placeholders (e.g., for criteria trees) and ensure every in-text citation corresponds to a reference entry.
>
> ---
>
> [Q1] Train–test splits
>
> Thank you for noting that this was unclear. We agree the dataset construction should be described more explicitly. Our pipeline uses three disjoint datasets drawn from two public sources.
>
> For Gold GenRM training, we use the human-labeled LitBench dataset. LitBench consists of (prompt, chosen_story, rejected_story) triplets labeled by humans. We sample 22k triplets, split into 21k for training and 1k for validation, and fine-tune Qwen3-32B to predict the human-preferred continuation.
>
> For Proxy GenRM training, we take a separate, non-overlapping set of 22k LitBench triplets and re-annotate chosen/rejected labels using the trained Gold evaluator. This yields 21k training and 1k validation examples for both answer-only and thinking GenRMs, used for training and static evaluation.
>
> For policy training, we use only prompts from the WritingPrompts dataset. We sample 91k prompts (90k for training, 1k for validation), ensuring there is no overlap with LitBench. All policy experiments use this set, except the β-sweeps (Figure 6) and the amount-of-GenRM-training experiments (Figure 2b), where we train on 40k prompts due to compute limits.
>
> In the revision, we will add a compact dataset and splits summary to the methodology section to make these distinctions explicit and avoid ambiguity.

---

### Meta-Review · Area_Chair_V1km · 2026-01-03

**Summary:**

This work investigates the scaling behavior of generative reward models for RLAIF when used as replacements for traditional Bradley–Terry models, across various scales of policy and reward models.

**Reviewer Concerns:**

The reviewers are mainly concerned about the following aspects:

1.	invalid or unfair comparison that cannot support the claimed conclusion

2.	unclear presentation, misleading phrasing, and unprecise explanations

3.	limited scope of domains, backbones, and optimization methods for evaluation

4.	lack of clarification and explanations about important details

Several reviewers unanimously raised some of these concerns. For example, Reviewers UekM, b7qy, and c46P all conveyed similar concerns about the invalid comparison and misleading statements. Despite the author responses in the rebuttal, the AC considers these critical issues still not fully resolved and in need of more comprehensive addressing in a future revision.

**Reviewer Scores:**

The reviewers' final scores would be 4, 4, 2, 8.

---

### Decision · Program_Chairs · 2026-01-26

Reject